# Computational design and interpretation of single-RNA translation experiments

**Luis U. Aguilera**[1], **William Raymond**[1,2], **Zachary R. Fox**[2], **Michael May**[2], **Elliot Djokic**[2], **Tatsuya Morisaki**[3], **Timothy J. Stasevich**[3,4], **Brian Munsky**[1,2]*

**1** Department of Chemical and Biological Engineering, Colorado State University Fort Collins, Colorado, United States of America, **2** School of Biomedical Engineering, Colorado State University Fort Collins, Colorado, United States of America, **3** Department of Biochemistry and Molecular Biology and Institute for Genome Architecture and Function, Colorado State University, Fort Collins, Colorado, United States of America, **4** Cell Biology Unit, Institute of Innovative Research, Tokyo Institute of Technology, Nagatsuta-cho 4259, Midori-ku, Yokohama, Kanagawa, Japan

* Brian.Munsky@colostate.edu

**Data Availability Statement:** The rSNAPsim simulation package is available at: https://github.com/MunskyGroup/rSNAPsim.git. All experimental data and codes used to fit models to that data are

## Abstract

Advances in fluorescence microscopy have introduced new assays to quantify live-cell translation dynamics at single-RNA resolution. We introduce a detailed, yet efficient sequence-based stochastic model that generates realistic synthetic data for several such assays, including Fluorescence Correlation Spectroscopy (FCS), ribosome Run-Off Assays (ROA) after Harringtonine application, and Fluorescence Recovery After Photobleaching (FRAP). We simulate these experiments under multiple imaging conditions and for thousands of human genes, and we evaluate through simulations which experiments are most likely to provide accurate estimates of elongation kinetics. Finding that FCS analyses are optimal for both short and long length genes, we integrate our model with experimental FCS data to capture the nascent protein statistics and temporal dynamics for three human genes: KDM5B, $\beta$-actin, and H2B. Finally, we introduce a new open-source software package, RNA Sequence to NAscent Protein Simulator (ʀSNAPsɪM), to easily simulate the single-molecule translation dynamics of any gene sequence for any of these assays and for different assumptions regarding synonymous codon usage, tRNA level modifications, or ribosome pauses. ʀSNAPsɪM is implemented in Python and is available at: https://github.com/MunskyGroup/rSNAPsim.git.

## Author summary

Translation is an essential step in which ribosomes decipher mRNA sequences to manufacture proteins. Recent advances in time-lapse fluorescence microscopy allow live-cell quantification of translation dynamics at the resolution of single mRNA molecules. Here, we develop a flexible computational framework to reproduce and interpret such experiments. We use this framework to explore how well different single-mRNA translation experiment designs would perform to estimate key translation parameters. We then integrate experimental data from the most flexible design with our stochastic model framework to reproduce the statistics and temporal dynamics of nascent protein elongation for

available at: https://github.com/MunskyGroup/Aguilera_PLoS_CompBio_2019.git.

**Funding:** LUA, WR, ZRF, MM, ED and BM were supported by the National Institute of General Medical Sciences of the National Institutes of Health under award number R35GM124747 and LUA, ZRF, MM, TM, TJS and BM were supported by the WM Keck Foundation. ZRF and MM were partially supported by a National Science Foundation grant (DGE-1450032). Any opinions, findings, conclusions or recommendations expressed are those of the authors and do not necessarily reflect the views of the funders. The funders had no role in study design, data collection and analysis, decision to publish, or preparation of the manuscript.

**Competing interests:** The authors have declared that no competing interests exist.

three different human genes. Our validated computational method is packaged with a simple graphical user interface that (1) starts with mRNA sequences, (2) generates discrete, codon-dependent translation models, (3) provides visualization of ribosome movement as trajectories or kymographs, and (4) allows the user to estimate how optical single-mRNA translation experiments would be affected by different genetic alterations (e.g., codon substitutions) or environmental perturbations (e.g., tRNA titrations or drug treatments).

## Introduction

The central dogma of molecular biology (i.e., DNA codes are transcribed into messenger RNA, which are then translated to build proteins) has been a foundation of biological understanding since it was stated by Francis Crick in 1958. Despite their overwhelming importance to biological and biomedical science, many of the fundamental steps in the gene expression process are only now becoming observable in living cells through the application of real time single-molecule fluorescence imaging approaches. Single-molecule imaging of transcription was first achieved two decades ago through the use of the MS2 system [1], which uses bacteriophage gene sequences to encode for specific and repeated stem-loop secondary structures in the transcribed mRNAs. These stem-loops are subsequently recognized and bound by multiple fluorescently-tagged MS2 Coat Proteins (MCP), which produce bright fluorescent spots that allow for the detection and spatial tracking of single-mRNA [2]. Tracking these labeled RNA has made it possible to observe many aspects of RNA dynamics that were previously obscured using bulk RNA measurements, such as the production of RNA from different alleles [3], the movement of mRNA–protein complexes from nucleus to cytoplasm through nuclear pores [4], and the association of RNA with different regions of the cell [5].

Even more recently, imaging single-molecule translation has also become possible through the discovery of similar approaches [6–10]. In this case, the mRNA is modified to encode for multiple epitopes in the open reading frame of a protein of interest (POI). As the protein is translated, these epitopes are quickly recognized and bound by fluorescent antibody fragment probes, Fig 1A. By combining the MS2 approach with these epitope recognition sites, the co-localization of mRNA spots and nascent protein spots reveal single-mRNA molecules that are undergoing active translation, as shown in Fig 1B. As was the case for single-RNA tracking, precise spatiotemporal imaging of translation sites within single living cells allows for multiple advances in comparison to bulk or single-cell assays [11]. For example, Morisaki and Stasevich recently reviewed three different approaches to track the translation dynamics for individual mRNA molecules over time and then use these data to infer translation rates [12]. The first design is related to Fluorescence Correlation Spectroscopy (FCS), in that the nascent protein fluorescence signal is monitored over time and used to compute the auto-covariance function (G($\tau$), Fig 1C, bottom). The time, $\tau_{FCS}$, at which G($\tau$) reaches zero denotes the characteristic time for a ribosome to translate the gene from the tag region to the end of the protein of interest [6]. A second approach to measure translation rate is to chemically block translation initiation (e.g., through the application of a drug such as Harringtonine, as depicted in Fig 1D, top). In this Run-Off Assay (ROA) approach, the time, $\tau_{ROA}$, at which the fluorescence signal disappears corresponds to the time for a single ribosome to translate the entire coding region, including the tag region itself [10]. A third technique, shown in Fig 1E, uses Fluorescence Recovery After Photobleaching (FRAP) to optically eliminate the nascent protein fluorescence associated with a single mRNA and then

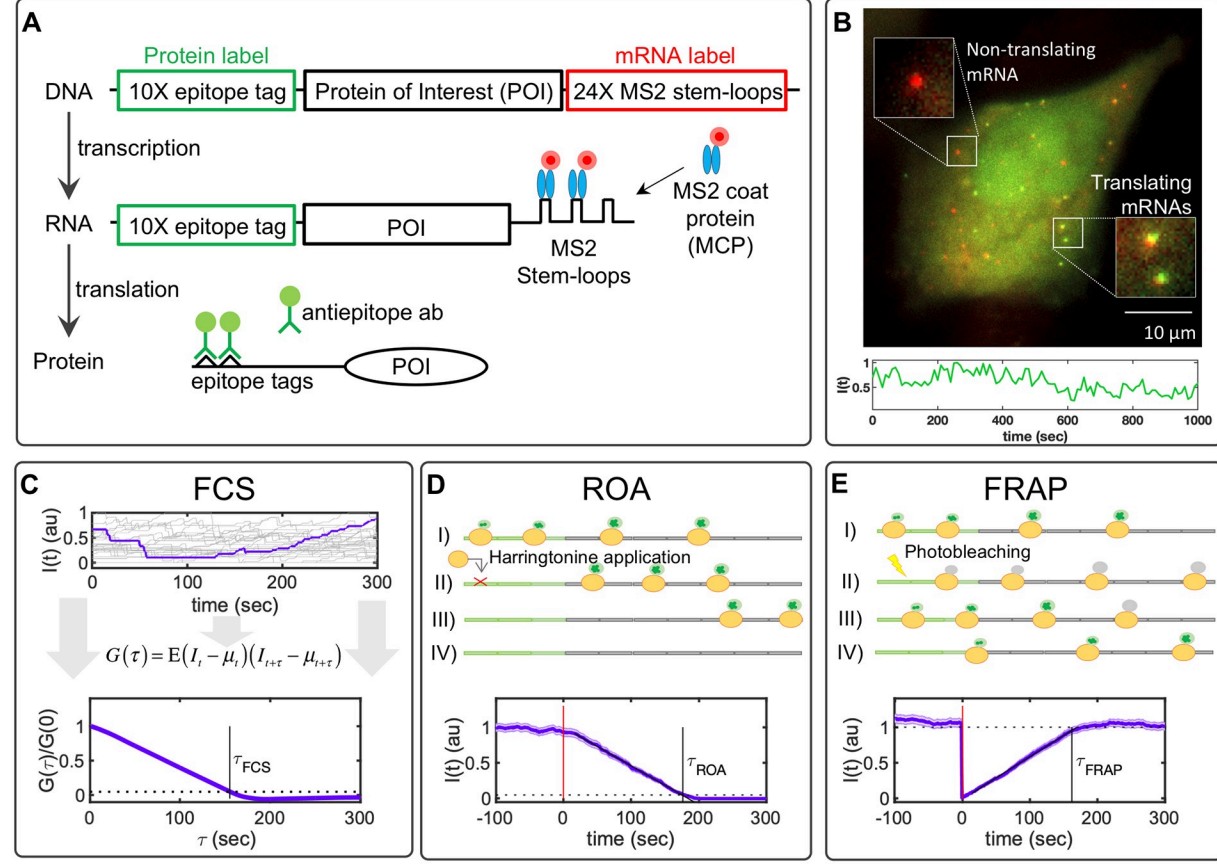

**Fig 1. Translation studies with single-molecule resolution.** A) Imaging single-molecule translation dynamics is achieved by the measurement of fluorescence spots that are produced when nascent proteins display epitopes that are recognized by antibody fragments bound to fluorescent probes. The gene construct encodes a 10X FLAG SM tag followed by a protein of interest (POI) and the 24X MS2 tag in the 3' UTR region. B) Microscopy image showing translation at single-molecule resolution; red spots represent single mRNA, and green spots represent nascent proteins. Below is a representative trace showing the intensity fluctuation dynamics of a single-transcript translating FLAG-10X-KDM5B. C) Simulated time courses representing the characteristic single-molecule fluctuation dynamics. A representative trace is selected and highlighted. At the bottom of the figure is given the normalized auto-covariance function ($G(\tau)$) calculated from simulated time courses. The time at which $G(\tau)$ hits zero represents the dwell-time ($\tau_{FCS}$). D) Harringtonine inhibits the translation initiation step by binding to the ribosomal 60S subunit. The plot shows the average fluorescence after Harringtonine treatment. Without new initiation events, the fluctuations diminish causing the intensity to drop to zero at time $\tau_{ROA}$. E) FRAP causes a rapid drop in the fluorescent intensity and a subsequent recovery that is proportional to the time needed by ribosomes to produce new nascent proteins with non-photobleached probes. The bottom plot shows the temporal dynamics of FRAP, where it can be observed by the abrupt decrease in intensity and a recovery time ($\tau_{FRAP}$) correlated with the gene length. All simulations correspond to KDM5B for 100 spots and a frame rate of 1 FPS. Error bars represent the standard error of the mean.

record the recovery of the signal to its original level. As for the FCS approach, the time of total recovery, $\tau_{FRAP}$, relates to the time required for a single ribosome to complete translation from the tag region to the termination codon [8, 9].

The temporal resolution offered by live single-mRNA, nascent translation imaging makes it possible to directly visualize and quantify initiation, elongation, and termination processes in live-cells [13]. Single-molecule studies have uncovered previously unknown events and mechanisms taking place during translation, such as the presence of active and inactive transcripts in specific locations in the cell [6, 9], different elongation rates caused by codon-optimized sequences [10], the spatiotemporal translation of specific genes in specific cellular compartments [7, 8], ribosomal frameshifting with bursty dynamics [14], and non-canonical forms of translation [15].

As these experimental techniques rapidly evolve, they induce a growing need for precise and flexible computational tools to interpret the resulting data and to design the next wave of single-RNA translation experiments. To help fill this gap, we present a versatile new set of computational design tools to estimate which specific single-mRNA translation dynamics experiments would provide the most accurate inference of model parameters. We demonstrate the generality of our analyses by simulating results for several different single-molecule experiments for a large database of human genes. We explore these different combinations of gene and experiment to ask which methodologies are better to measure specific biophysical parameters and for which types of genes. We then constrain our model by fitting it to experimental data for several genes. Finally, we describe and demonstrate the use of a new open-source and user-friendly software package: RNA Sequence to NAscent Protein Simulation (RSNAPSIM), which allows the user to easily simulate the single-molecule translation dynamics of any gene. Finally, we discuss future directions and the potential limitations of the current form of this new technology.

## Modeling single-RNA translation dynamics

To simulate translation with single-molecule resolution, we adopted a stochastic model of polymerization that is similar to those developed previously in [16–18]. We then extend this model to allow for variable ribosome sizes, codon- and tRNA-dependent translation elongation rates, and arbitrary placement of fluorescent probe binding epitopes, and we analyze these models specifically in the context of single-mRNA translation as observed using time-lapse fluorescence microscopy experiments (e.g., Fig 1).

The model consists of a set of reactions where random variables $\{x_i\}$ represent the fluctuating occupancy of ribosomes at each specific $i^{\text{th}}$ codon along a single mRNA,

$$\varnothing \xrightarrow{w_0(x_1,\ldots,x_{nf})} x_1 \xrightarrow{w_1(x_1,\ldots,x_{nf+1})} \ldots x_i \xrightarrow{w_i(x_i,\ldots,x_{nf+i})} \ldots \xrightarrow{w_t} x_L, \tag{1}$$

where $L$ is the length of the gene in codons, $n_f$ is the ribosome footprint, and $\mathbf{x} = [x_1, x_2, \ldots, x_L] \in \mathbb{B}^L$ is a binary vector of zeros and ones, known as the occupancy vector, which represents the presence ($x_i = 1$) or absence ($x_i = 0$) of ribosomes at every $i^{\text{th}}$ codon. The initial reaction in the model describes the initiation step, where the ribosomes bind to the mRNA at the rate $w_0(x_1, \ldots, x_{nf})$. Ribosomes are large biomolecules that occupy around 20 to 30 nuclear bases (or seven to 10 codons) once bound to the mRNA [19]. This is captured in the model by specifying the ribosome footprint, $n_f = 9$, which guarantees that initiation cannot occur if another downstream ribosome is already present within the first $n_f$ codons, Fig 2A. This binding restriction can be written simply as:

$$w_0 = k_i \prod_{j=1}^{nf} (1 - x_j), \tag{2}$$

where $k_i$ is the initiation constant, and the product is equal to one if and only if there are no ribosomes within the first $n_f$ codons.

Similarly, we represent the elongation reactions, where the ribosome moves along the mRNA from codon to codon in direction 5' to 3' according to:

$$w_i = k_e(i) \cdot x_i \prod_{j=1}^{nf} (1 - x_{i+j}), \quad \text{for } i = 1, \ldots, L-1; \tag{3}$$

where $k_e(i)$ is the elongation rate at the $i^{\text{th}}$ codon, and the product again enforces ribosome

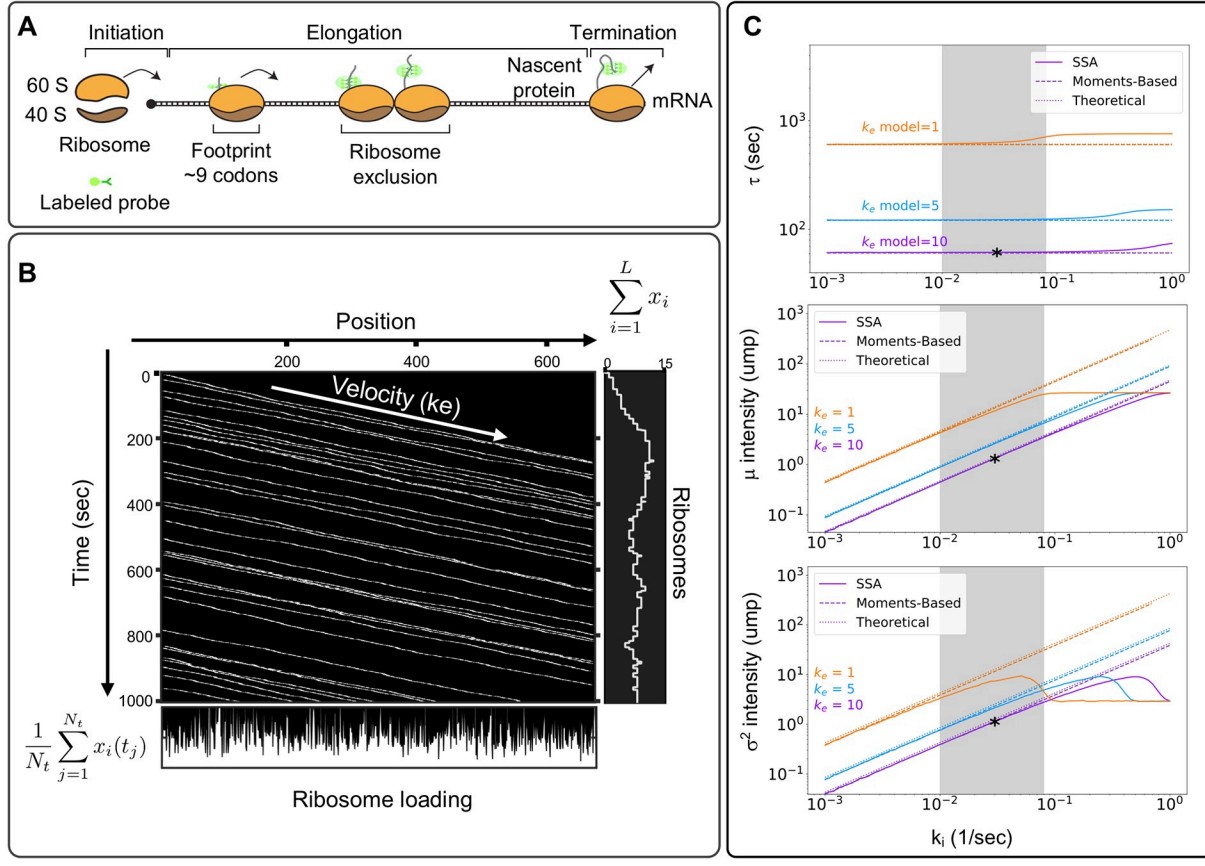

**Fig 2. Modeling single-molecule translation.** A) Translation is divided into three main processes: initiation, elongation, and termination. The ribosome footprint represents the physical space occluded by the ribosome, enforcing that no two ribosomes can occupy the same space and time. B) Kymographs represent ribosome movement as a function of time (y-axis) and position (x-axis). Each line represents a single ribosome trajectory. The average slope is proportional to the effective ribosome elongation rate. The plot to the right shows the relationship between ribosome movement and fluorescence intensity, and the plot below shows the ribosome loading at each codon position, calculated as the time-average of ribosome occupancy at the corresponding codon. C) Comparison of the average elongation time (top) and the mean (middle) or variance (bottom) of fluorescence intensity as calculated using the simplified model (Eqs 18 to 21), a linear moments-based model (Eqs 9 to 17), and a full stochastic model (Eqs 1 to 5). Gray area represents previously reported parameter values for ribosome initiation. Panels B and C correspond to simulations for the β-actin. Asterisks represent the specific parameter combination used for Table 1.

exclusion. To implement the effect of codon-usage bias and tRNA availability during protein synthesis, we adopt a similar argument to that presented by Georgoni et al., [16]: rare codons are correlated with low tRNA abundance, which cause a longer waiting time for the ribosome to synthesize the given amino acid at that codon. As tRNA concentrations have been related to codon usage [20], we assume each codon's elongation rate is proportional to its usage in the human genome according to:

$$k_e(i) = \bar{k}_e \cdot (u(i)/\bar{u}), \tag{4}$$

where $u(i)$ denotes the codon usage frequency in the human genome (given in S1 Table from [21]), $\bar{u}$ represents the average codon usage frequency in the human genome, and the global parameter $\bar{k}_e$ is an average elongation constant, which can be determined through experiments.

Although simple in its specification, the above model allows for many adjustments to explore different experimental circumstances. As a few examples, (i) one can represent translation

inhibition analyses such as those performed in [7] by making the initiation rate, $k_i$, a function of time or external input; (*ii*) one can analyze effects of synonymous codon substitution by replacing codons with their more or less common relatives; (*iii*) one can represent codon depletion, as studied in [16] by reducing the corresponding rates $k_e(i)$ for all $i$ corresponding to the depleted tRNA; (*iv*) one could explore the effects of pausing or traffic jams at specific codons by reducing $k_e(i)$ at specific codons, or (*v*) one can represent bursting kinetics by replacing the constant $k_i$ with a discrete-stochastic activation/deactivation process. We will explore several of these circumstances below.

**Kymograph representation of single-mRNA translation dynamics.** With our simple specification of the translation initiation, elongation and termination reactions, we can now simulate random trajectories, $\mathbf{x}(t)$, which we collect to form binary occupancy trajectory matrices $\mathbf{X} = [\mathbf{x}(t_1)^T, \ldots, \mathbf{x}(t_{N_t})^T]^T \in \mathbb{B}^{N_t \times L}$, where each row refers to the $i^{\text{th}}$ position on the gene, and each column represents a specific time $t_j$. To visualize ribosome movement trajectories, each random $\mathbf{X}$ can be plotted in two dimensions (position v.s. time) to form kymographs similar to those extensively used to represent organelle movement [22]. For example, Fig 2B shows a visualization of $\mathbf{X}$ for a case study on the $\beta$–actin gene. Each line from left to right on the kymograph corresponds to the movement of a single ribosome from initiation to termination. We note that averaging along the columns of $\mathbf{X}$ (i.e., in the vertical direction of the kymograph) yields the time-averaged loading of the ribosomes at each codon position, and summing across the rows of $\mathbf{X}$ (i.e., in the horizontal direction of the kymograph) yields the number of ribosomes for that mRNA at each instant in time.

**Relating protein elongation dynamics to fluorescence signal intensities.** To relate our model describing ribosome occupancy to experimental measurements of translation spot fluorescence, we introduce a *fluorescence intensity vector* that converts the instantaneous occupancy vector, $\mathbf{x}(t)$, to the total number of translated epitopes available to bind to fluorescent markers. This intensity vector can be written as:

$$I(t) = \sum_{i=1}^{L} c_i \cdot x_i(t) = \mathbf{c}\mathbf{x}^{\text{T}}, \tag{5}$$

where $\mathbf{c} = [c_1, c_2, \ldots, c_L]$ and each $c_i$ is the cumulative number of fluorescent probes bound to epitopes encoded at positions $(1, \ldots, i)$ along the mRNA. For example, $\mathbf{c}$ would be defined as $\mathbf{c} = [0, 0, 1, 1, 2, 2, 3, 3, \ldots, 3]$ for an RNA sequence with epitopes encoded at positions [3, 5, 7]. We note that the random occupancy matrices, $\mathbf{X}$, are easily converted to intensity time traces using the simple algebraic operation $\mathbf{I} = [I(t_1), \ldots, I(t_{N_t})] = \mathbf{c}\mathbf{X}^{\text{T}}$.

## Simplifications for combinatorial analyses of genes, parameters, and experiment designs

The model as defined above is sufficient to simulate fluorescence dynamics for any specified gene and for a vast range of potential time-lapse microscopy experiments. However, these simulations become computationally intensive when studying combinations of thousands of genes, using thousands of different parameters sets, and for hundreds of different experiment designs. To ameliorate this concern, we next introduce model simplifications that progressively remove elements from the original model, such as ribosome exclusion and single-codon resolution, while retaining effects of codon-dependent translation rates and the geometric placement of fluorescent tags. We then test under what conditions (i.e., parameters and gene lengths) these simplifications are valid, and we compare these conditions to experimentally reported values.

**Approximations for the means, variances, and auto-covariances of nascent translation kinetics.** When ribosome loading is sparse (e.g., for slow initiation or fast elongation such that ($k_i/ke \ll 1/nf$)), ribosome collisions will become negligible, and the nonlinearities in Eqs 2 and 3 have less effect on the overall ribosome dynamics. Under such circumstances, it is possible to derive a simplified linear system model for the elongation dynamics. In the linear model, the propensity of the codon-dependent elongation step (Eq 2) is simplified to $w_i(x_i) = k_i x_i$ such that the ability of a ribosome to add another amino acid only depends on the current position of the ribosome, and not on the footprint of other ribosomes.

We define the reaction stoichiometry matrix to describe the change in the ribosome loading vector, $\mathbf{x}$, for every reaction as:

$$\mathbf{S}_{i,j} = \begin{cases} 1 & \text{for all } i = j, \\ -1 & \text{for all } i = j - 1, \end{cases} \tag{6}$$

where $i$ corresponds to each codon in the protein of interest. The first column of $\mathbf{S}$ corresponds to the initiation reaction, the next $L - 1$ columns refer to elongation steps when an individual ribosome transitions from the $i^{\text{th}}$ to the $i + 1^{\text{th}}$ codon, and the final column corresponds to the final elongation step and termination. Maintaining the same order of reactions, and neglecting ribosome exclusion, the propensities of all reactions can be written in the affine linear form as:

$$\mathbf{w} = \mathbf{w}_0 + \mathbf{W}_1 \mathbf{x}, \tag{7}$$

where $\mathbf{w}_0$ is a column vector of zeros with the first entry $k_i$, and $\mathbf{W}_1$ is a matrix defined as:

$$[\mathbf{W}_1]_{i,j} = \begin{cases} k_e(i) & \text{for all } i = j + 1, \\ 0 & \text{otherwise .} \end{cases} \tag{8}$$

Using the definition of the fluorescence intensity from Eq 5, the first two uncentered moments of the intensity $I(t)$ can be written in terms of the ribosome position vector $\mathbf{x}(t)$ as:

$$\mathbb{E}\{I(t)\} = \mathbb{E}\{\mathbf{c}\mathbf{x}(t)\} = \mathbf{c}\mathbb{E}\{\mathbf{x}(t)\}, \tag{9}$$

$$\Sigma_I(0) = \mathbb{E}\{(I(t) - \mathbb{E}\{I(t)\})^2\} = \mathbf{c}\Sigma_{\mathbf{x}}(0)\mathbf{c}^{\text{T}}, \tag{10}$$

where $\mathbb{E}\{\mathbf{x}(t)\}$ and $\Sigma_{\mathbf{x}}(0)$ are the mean and zero-lag-time variance in the ribosome occupancy vector, respectively. For the approximate linear propensity functions in Eq 7, the moments of the ribosome position vector are governed by the equations [23]:

$$\frac{d\mathbb{E}\{\mathbf{x}\}}{dt} = \mathbf{S}\mathbf{W}_1\mathbb{E}\{\mathbf{x}\} + \mathbf{S}\mathbf{w}_0 \tag{11}$$

$$\frac{d\Sigma_{\mathbf{x}}}{dt} = \mathbf{S}\mathbf{W}_1\Sigma_{\mathbf{x}} + \Sigma_{\mathbf{x}}\mathbf{W}_1^T\mathbf{S}^T + \mathbf{S}\text{diag}(\mathbf{W}_1\mathbb{E}\{\mathbf{x}\} + \mathbf{w}_0)\mathbf{S}^T. \tag{12}$$

By setting the left hand side of Eq 11 to zero, the steady-state mean ribosome loading vector can be found by solving the algebraic expression:

$$\mathbf{S}\mathbf{W}_1\mathbb{E}\{\mathbf{x}\} + \mathbf{S}\mathbf{w}_0 = 0. \tag{13}$$

Similarly, the steady-state covariance matrix, $\Sigma_{\mathbf{x}}$, in the ribosome loading vector is given by the

solution to the Lyapunov equation (from right hand side of Eq 12):

$$\mathbf{SW}_1\mathbf{\Sigma}_\mathbf{x} + \mathbf{\Sigma}_\mathbf{x}\mathbf{W}_1^T\mathbf{S}^T + \mathbf{S}\text{diag}(\mathbf{W}_1\mathbb{E}\{\mathbf{x}\} + \mathbf{w}_0)\mathbf{S}^T = 0. \tag{14}$$

The auto-covariance dynamics of the nascent protein fluorescence intensity is defined:

$$\begin{aligned} G(\tau) &= \mathbb{E}\{(I(t) - \mathbb{E}\{I(t)\})(I(t+\tau) - \mathbb{E}\{I(t+\tau)\})\} \\ &= \mathbb{E}\{\mathbf{c}(\mathbf{x}(t) - \mathbb{E}\{\mathbf{x}(t)\})(\mathbf{x}(t+\tau) - \mathbb{E}\{\mathbf{x}(t+\tau)\})^T\mathbf{c}^T\} \\ &= \mathbf{c}\mathbb{E}\{(\mathbf{x}(t) - \mathbb{E}\{\mathbf{x}(t)\})(\mathbf{x}(t+\tau) - \mathbb{E}\{\mathbf{x}(t+\tau)\})^T\}\mathbf{c}^T \end{aligned} \tag{15}$$

$$= \mathbf{c}\mathbf{\Sigma}_\mathbf{x}(\tau)\mathbf{c}^T, \tag{16}$$

where $\mathbf{\Sigma}_\mathbf{x}(\tau)$ is the cross-covariance of the ribosome occupancies at a lag time of length $\tau$. Noting that the probe design, $\mathbf{c}$, is constant with respect to $\tau$, it is only necessary to find the cross-covariances of the ribosome occupancy. Following the regression theorem [24], these covariances are given by the solution to the set of ODEs,

$$\frac{d\mathbf{\Sigma}_\mathbf{x}(\tau)}{d\tau} = \phi\mathbf{\Sigma}_\mathbf{x}(\tau), \tag{17}$$

where the initial condition is provided by steady-state covariance (i.e., the solution for $\mathbf{\Sigma}_\mathbf{x}(0)$ in Eq 14) and the autonomous matrix of the process is given by $\phi = \mathbf{SW}_1$. Integrating Eq 17, the auto-covariance of the intensity $G(\tau)$ can be found using Eq 16. We reiterate the fact that this simplification relies only on the assumption of sparse loading of ribosomes on the mRNA, and the moments analyses in Eqs 13, 14 and 17 retain the codon-dependent rate through the definition of the matrix $\mathbf{W}_1$ and the specific positions of probes through the definition of the vector $\mathbf{c}$.

**Simplified algebraic expressions for nascent translation kinetics.** In the limit of low initiation events and long genes, the probe region can be further approximated by a single point, and the above model can be simplified even further to allow direct estimation of steady-state translation features. First, since the average time for a ribosome to move one codon is $\mathbb{E}\{\Delta t_i\} = 1/k_e(i)$, the total average time it takes a ribosome to complete translation from the start codon to the end of the mRNA is:

$$\mathbb{E}\{\tau\} = \sum_{i=1}^{L}\frac{1}{k_e(i)}, \tag{18}$$

where $L$ is the gene length. Using the codon-dependent translation rates from Eq 4, we can modify Eq 18 to

$$\mathbb{E}\{\tau\} = \frac{1}{\bar{k}_e}\sum_{i=1}^{L}\frac{\bar{u}}{u(i)}. \tag{19}$$

If one could experimentally measure $\tau_{\text{Exp}}$ using one of the techniques described above, then $\bar{k}_e$ could be estimated as:

$$\bar{k}_e \approx \frac{1}{\tau_{\text{Exp}}}\sum_{i=n_p}^{L}\frac{\bar{u}}{u(i)}, \tag{20}$$

where $n_p$ is the effective codon position of the fluorescent tag. In practice, the specification of $n_p$ will vary depending upon the type of experiment (e.g., FCS, FRAP or ROA) used to estimate $\tau_{\text{Exp}}$, as will be discussed in more detail below.

Given the apparent association time of a ribosome on the mRNA ($\tau$) and the initiation rate ($k_i$), the distribution for the number of visible ribosomes on a transcript at steady state can also be estimated using this simplified model. Under the assumption that each initiation event is an independent and exponentially distributed random event, the number of ribosomes downstream from the $n_b^{\text{th}}$ codon, and therefore the fluorescence in units of mature proteins, would be approximated by a Poisson distribution with mean (and variance) equal to

$$\mu \approx \sigma^2 \approx k_i \cdot \tau. \tag{21}$$

For a more realistic treatment of the fluorescence intensity, one could assume that the multiple probes are spread uniformly over a finite region, such that the fluorescence will increase linearly as ribosomes pass through the probe region. To approximate this gradual increase in fluorescence, Eq 21 can be corrected by a multiplicative factor (see Methods) as:

$$\mu_I \approx k_i \cdot \tau \left( 1 - \frac{L_t}{2L} \right), \tag{22}$$

$$\sigma_I^2 \approx k_i \cdot \tau \left( 1 - 2\frac{L_t}{3L} \right), \tag{23}$$

where $L_t$ is the length of the tag region (e.g., $L_t$ = 318 aa for the 10X FLAG 'Spaghetti Monster' SM-tag used in [6]).

**Agreement of full and simplified models for codon-dependent translation kinetics.** To demonstrate the close agreement between the full stochastic model, the reduced linear moments model, and the simplified theoretical analysis, Table 1 compares the model generated values for each of the quantities $\tau$, $\mu_I$, and $\sigma_I^2$ for three different human genes H2B ($L$ = 128aa), $\beta$-actin ($L$ = 375aa), and KDM5B ($L$ = 1549aa), using reported parameters of $k_i$ = 0.03 s$^{-1}$ and $\bar{k}_e = 10$ s$^{-1}$ [6]. For further comparison, Fig 2C compares estimates of $\tau$ (top), $\mu_I$ (middle), and $\sigma_I^2$ (bottom) for the $\beta$-actin gene for each of the three analyses, and as a function of different initiation and elongation rates. This comparison demonstrates that, at least for fast elongation rates, the full stochastic analysis and the moments-based computation are in excellent agreement to estimate the effective time as well as the mean and variance in the level of nascent proteins per RNA. However, when the initiation rate approaches $\bar{k}_e/nf$, ribosome collisions become more prevalent, which substantially lengthens the effective elongation time (Fig 2C top), and leads to a saturation of ribosomes (Fig 2C middle and bottom), and these nonlinear behaviors are not captured by the moment-based model. For longer genes, the simplified theoretical estimates from Eqs 18–21 are also in good agreement with the complete model. For shorter genes, it becomes less realistic to approximate the tag region with a single point or a uniform distribution, and the error of this approximation leads to poorer estimates of the elongation time and the Poisson approximation over-estimates the true variance (see H2B in Table 1). However, even for short genes, the linear moments-based model, which includes the exact positions of all probes and the codon usage, provides a more accurate estimate of the true system behaviors.

## Results

Having demonstrated close agreement of the simplified theoretical models with the full stochastic simulations, we can now use the much more computationally efficient theoretical analyses to explore how well different experiment designs should be expected to estimate translation parameters from single-RNA translation dynamics.

**Table 1. Comparing model dynamics.**

| | Stochastic Model | Moments-Based Model | Theoretical Model |
|---|---|---|---|
| *KDM5B*, L = 1867aa | | | |
| mean ($\mu$) | 5.2 ± 0.02 | 5.07 | 5.01 |
| var ($\sigma^2$) | 4.5 ± 0.03 | 4.90 | 4.93 |
| period ($\tau$) | 187.78 ± 0.94 | 180.0 | 185.23 |
| $\beta - actin$, L = 693aa | | | |
| mean ($\mu$) | 1.31 ± 0.004 | 1.34 | 1.42 |
| var ($\sigma^2$) | 1.08 ± 0.003 | 1.17 | 1.28 |
| period ($\tau$) | 62.17 ± 0.27 | 60.0 | 60.94 |
| *H2B*, L = 446aa | | | |
| mean ($\mu$) | 0.75 ± 0.002 | 0.77 | 0.82 |
| var ($\sigma^2$) | 0.56 ± 0.001 | 0.59 | 0.68 |
| period ($\tau$) | 42.96 ± 0.07 | 42.0 | 41.80 |

Mean and variance of intensity are given in units of mature proteins (ump). The period ($\tau$) has units of seconds. Elongation and initiation rates are $\bar{k}_e = 10s^{-1}$ and $\bar{k}_i = 0.03s^{-1}$, respectively. Lengths include the tag region of 318aa. Stochastic simulations were performed for 500 simulated spots, with a frame rate of 1 sec, and for 2,000 frames. Error values represent the standard deviation of 3 repetitions of independent simulations.

## Design of experimental assays for improved quantification of translation kinetics

Using the models above, and if we could experimentally estimate the average time that ribosomes take to translate a single complete protein from a given gene, $\tau^{(g)}$, we could estimate $\bar{k}_e^{(g)}$ using Eq 20. With this in mind, we next consider three approaches that have been used to estimate $\tau^{(g)}$ in recent experimental investigations (Fig 1C–1E): Fluorescence Correlation Spectroscopy (FCS), Run-Off Assays (ROA), and Fluorescence Recovery After Photobleaching (FRAP). Using our full stochastic models to generate synthetic data and the simplified theoretical model to interpret these data, we ask how accurately would each of these three assays work to identify $\bar{k}_e^{(g)}$ for a comprehensive list of 2,647 human genes from the PANTHER database [25] and under different imaging conditions corresponding to different frame rates or numbers of mRNA spots.

In the FCS approach, we compute the auto-covariance function, $G(\tau)$ (defined in Eq 15), of the simulated fluorescence intensities, and from $G(\tau)$ we estimate the time lag, $\tau_{FCS}$, at which correlations disappear (see Fig 1C and Methods). In the ROA approach, we simulate the addition of a chemical compound, such as Harringtonine, which binds the 60S ribosome subunit and prevents ribosome assembly [26], and we record the average time, $\tau_{ROA}$, at which protein fluorescence disappears from the RNA (see Fig 1D and Methods). To approximate variability in the specific time at which the drug reaches the mRNA and blocks ribosome initiation, we assume that the time of initiation blockage occurs at a normally distributed time of 60 ± 10 seconds [27]. In the FRAP analysis, we simulate an instantaneous fluorescence bleaching of all nascent proteins and then record the average time, $\tau_{FRAP}$, at which fluorescence recovers to the average steady-state level, Fig 1E [28]. To reduce the effects of stochastic sample variation in these calculations, we applied a linear fit to ROA and FRAP experiments and determined $\tau_{ROA}$ and $\tau_{FRAP}$ when these intensities intersect defined thresholds of zero intensity for ROA or the mean recovered intensity for FRAP. For FCS, we estimate $\tau_{05}$ as the time the auto-covariance function drops below 5% of the zero-lag covariance and calculate $\tau_{FCS} = \tau_{05}/0.95$.

The specific location of probes along the mRNA has different effects on the fluorescence kinetics for the three experimental analyses. The characteristic decorrelation time in FCS and recovery time in FRAP are both set by the time it takes a single ribosome to translate from the tag region to the end of the mRNA. To reflect this, we define the approximate probe location, $np_{\text{FCS}}$ or $np_{\text{FRAP}}$ in Eq 20, as the beginning of the tag region. In this case, the beginning of the tag region is at the beginning of the gene, but in general, we note that moving the fluorescent tag regions downstream toward the 3' end would shorten the effective times measured using FCS or FRAP. In contrast, for the ROA, the characteristic time is defined by how long it takes from when translation initiation is blocked until all ribosomes complete translation. Because this time depends solely on the gene length, and not on the probe placement, we assume $np_{\text{ROA}} = 1$, independent of probe placement. In addition to these effects on average experiment timescale estimates, we note that placing probes as near as possible to the 5' end of the mRNA or using longer proteins increases the fluorescence signal-to-noise ratio for all three approaches and can reduce estimation uncertainties.

To generate simulated data, we assumed that all 2,647 genes in the library have a global average translation rate of $\bar{k}_e = 10$ sec$^{-1}$ and an initiation rate of $k_i = 0.03$ sec$^{-1}$. For each experiment type and each gene, we simulated time lapse microscopy data for 100 independent RNA and for 300 frames at 1/3 frames per second (FPS). We then estimated $\tau^{(g)}$ from these simulations using each of the three experimental methodologies, and we estimated the corresponding average elongation rate using the specific gene sequence and Eq 4. Under these conditions, Fig 3A–3C (top) show the resulting distributions of estimated $\tilde{k}_e$ for long genes (> 1000 codons, n = 658, purple), medium length genes (500 − 1000 codons, n = 1719, blue), and short genes (<500 codons, n = 270, orange) using each of the three experimental approaches. When all genes were analyzed at the same imaging conditions (100 spots, 300 frames, 1/3 FPS), the FCS approach was the most accurate with root mean squared (RMSE) of 0.63, 1.35, and 1.60 for short, medium and long genes, respectively. For comparison, ROA had RMSE of 2.22, 2.52, and 1.78 and FRAP had RMSE of 5.22, 4.58, and 2.68 for the same combinations of genes and imaging conditions.

We next extended our analysis to consider different numbers of spots and different frame rates at which to collect the data, but under the assumption that the total number of frames would remain fixed at 300. Fig 3A shows the corresponding resulting RMSE for different combinations of these experiment designs. As expected, we found the sampling rate and number of mRNA spots to directly affect the estimated $k_e^{(FCS)}$. FCS was the only technique capable to estimate the true elongation rate within a $RMSE_{\text{FCS}} \leq 2.0$ sec$^{-1}$ for short, medium and long genes. For short genes, this could be accomplished with as few as 10 spots with a frame rate of 1/3 FPS. Medium length and long genes could also be accurately quantified with 10 spots at frame rates of 1/3 FPS or 1/10 FPS.

The ROA was also capable to estimate the elongation rate to an accuracy of $RMSE_{\text{ROA}} < 2.0$ sec$^{-1}$ for medium and long genes, and for fast frame rates, the ROA approach could be more accurate than FCS. However, when applying the ROA method to short genes, we obtained $RMSE_{\text{ROA}} > 2.0$ sec$^{-1}$ under all combinations of sampling rates and repetition numbers at 100 or fewer spots (Fig 3B). This effect can be explained in that the number of ribosomes actively translating each mRNA is small and highly susceptible to stochastic effects in the case of small genes. We also note that the error using ROA depends strongly on the precision of the estimate for the specific time at which translation is blocked after application of Harringtonine; if the average value of this time is unknown, or if variations exceed our assumed standard deviation of 10 seconds, then accuracy using ROA is severely diminished, especially for short genes.

We found that FRAP substantially overestimates the elongation rates for short size genes, which can be observed in Fig 3C, where it is shown that recovering a $RMSE_{\text{FRAP}} < 2.0$ sec$^{-1}$

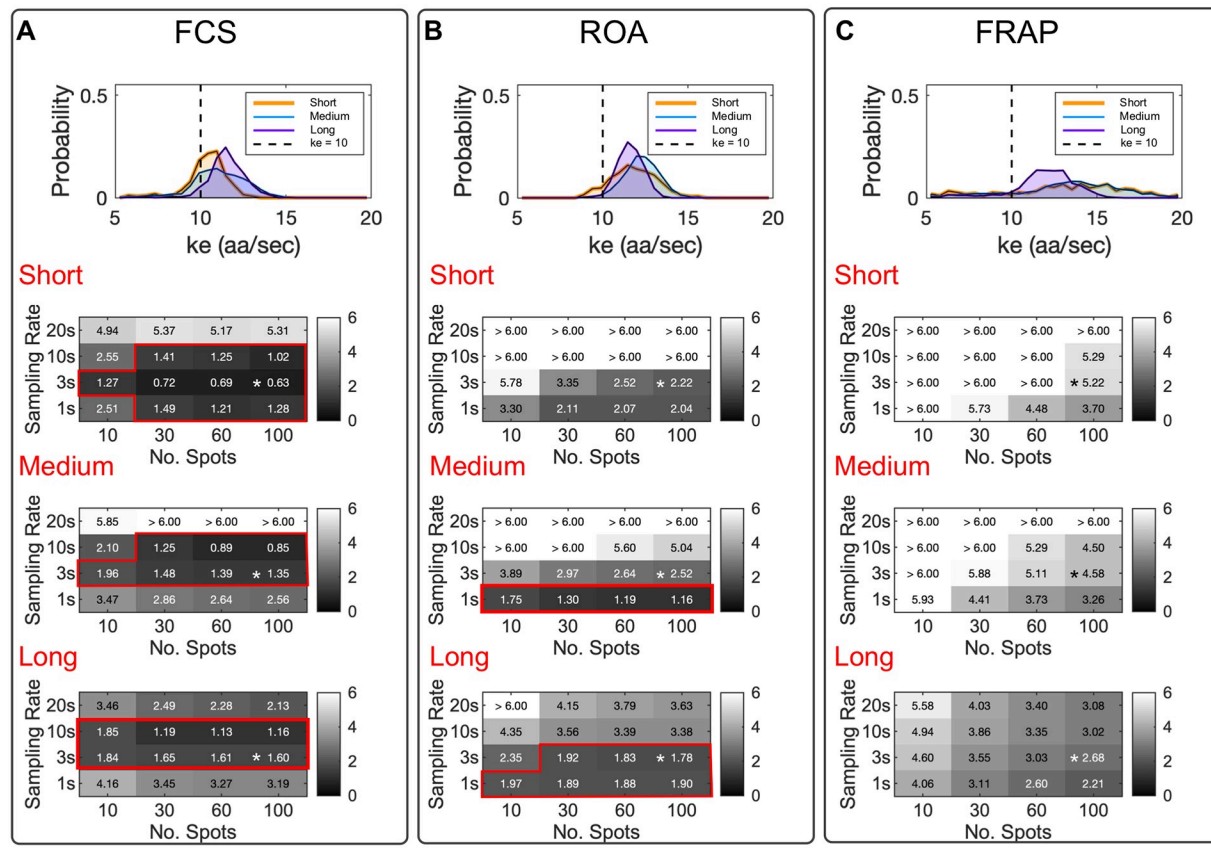

**Fig 3. Comparing experimental methodologies to estimate ribosome elongation rates.** Elongation rate estimate experiments were simulated for 2,647 human genes, using (A) Fluorescence Correlation Spectroscopy (FCS), (B) Run-Off Assays (ROA), and (C) Fluorescence Recovery After Photobleaching (FRAP). Top panels show the distributions of estimated $\tilde{k}_e$ for long genes ($> 1000$ codons, n = 658, purple), medium length genes ($500 - 1000$ codons, n = 1719, blue), and short genes ($< 500$ codons, n = 270, orange) using 100 mRNA spots for 300 frames at 1/3 FPS. The true elongation rate is denoted by a vertical dashed line. Bottom panels show the RMSE in elongation rate estimation as a function of the number of mRNA spots and the sampling rate. Red boxes highlight all experimental designs that yield a RMSE $< 2.0$. Asterisks represent the frame rate and number of repetitions used in panel (A). The 'true' elongation rate was set at $\bar{k}_e = 10$, and the initiation rate was fixed at $k_i = 0.03$ sec$^{-1}$ for all simulations.

was not possible for any of the considered combinations of the number of RNA spots and sampling rates. We argue that the estimate of elongation rates using FRAP is limited by the intrinsic formulation of the fluorescent probe design. FRAP requires an intensity generating mechanism to reestablish the fluorescence to a pre-perturbation steady state. For single-molecule translation studies, this mechanism relies on ribosomal initiation events that are rare and highly susceptible to variability [6–9]. This variability is reflected in the estimated $\tau_{\text{FRAP}}$ and in the final estimated elongation rate. Even for the more favorable medium and long length genes, our results indicate that for FRAP, a large number of mRNA spots ($>100$ mRNA spots) would be needed to achieve accurate estimates (Fig 3C).

## Calibration of the stochastic translation model using quantitative data from single-RNA translation experiments

Having determined that the FCS approach provides the most consistent estimate of elongation rate for genes of different lengths, we next turn to published experimental FCS data that quantified the fluctuation dynamics for three human gene constructs of different lengths: KDM5B

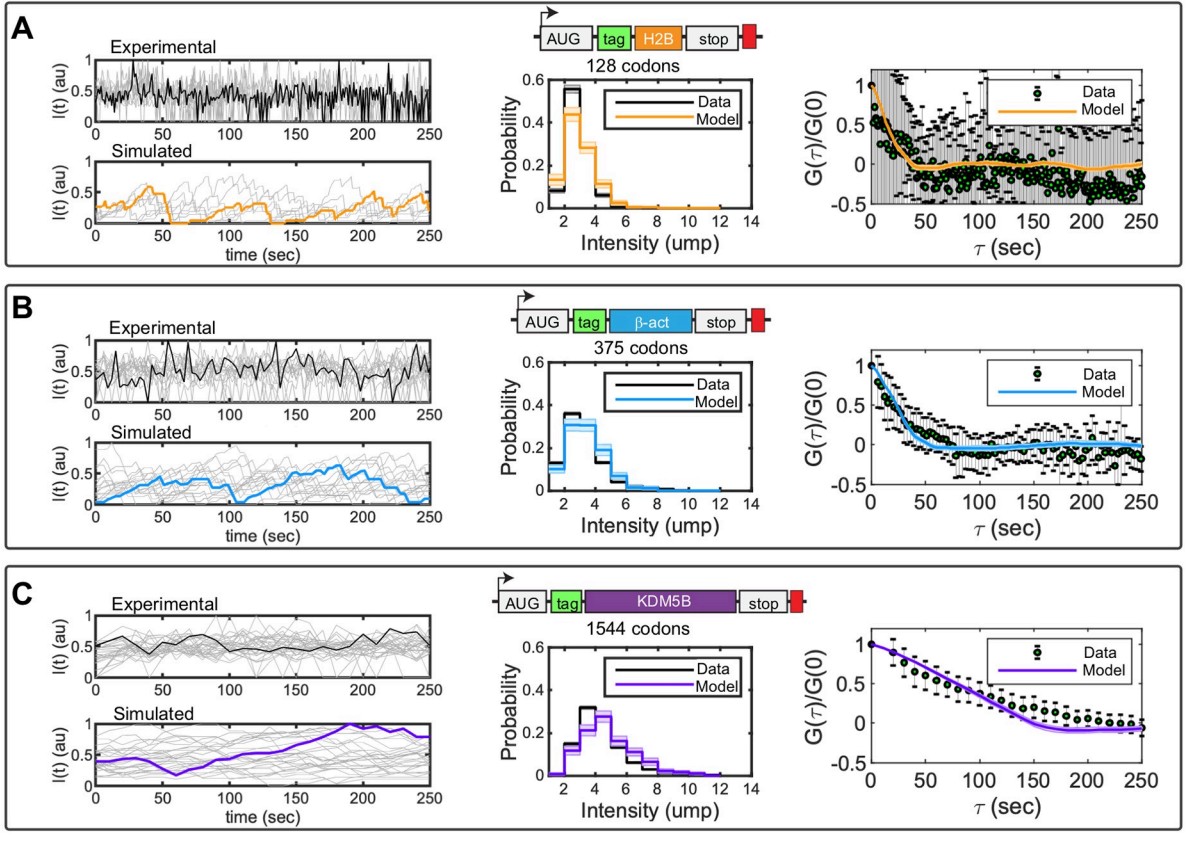

**Fig 4. Fitting single-molecule data with the full stochastic model.** Experimental data show the fluctuation dynamics of gene constructs encoding an N-terminal 10X FLAG 'Spaghetti Monster' SM-tag (green) followed by a protein of interest and finally a 24X MS2 tag (red) in the 3' UTR region. Three proteins were studied: A) H2B (orange), B) $\beta$-actin (blue) and C) KDM5B (violet). Middle figures show the simulated (colors) and measured (black) probability distributions for an mRNA to have a fluorescence intensity corresponding to $i$ units of mature proteins (ump). Right images show the normalized auto-covariance function ($G$) calculated from experimentally measured (black error bars) and computationally simulated (colors) autocorrelation functions. Error bars in the experimental data and shadow bars in the simulated auto-covariance plots represent the standard errors of the mean. Elongation and initiation rates were obtained by parameter optimization, using the Hooke and Jeeves Algorithm ([29]). Optimized parameters and their uncertainties (see Methods) are provided in Eq 29.

(1549 aa), $\beta$-actin (375 aa), and H2B (128 aa) [6]. Each construct encodes for an N-terminal 10X FLAG 'Spaghetti Monster' SM-tag (318 aa) followed by the specific protein of interest (POI), and the stop codon for each POI was followed by 24 repetitions of the MS2 tag in the 3' UTR region. For each construct, the MS2 signal was used to track the mRNA motion in three dimensions, and the co-localized fluorescence intensity of the FLAG SM-tag was quantified as a function of time. These movies were collected using frame rates of 1 sec for H2B (n = 10), 3 sec for $\beta$-actin (n = 17), and 10 sec for KDM5B (n = 35), and each trajectory was tracked for up to 300 frames per mRNA. Fig 4A–4C (left) show example time traces (in arbitrary units of fluorescence) for the nascent protein level per individual mRNA for each of the three genes. To achieve long trajectories, it is necessary to use low laser power, which introduces higher variability in signal intensities from one spot to another. Therefore, to account for variability in imaging settings between tracking experiments, all trajectories were normalized to have a variance of one prior to auto-covariance analysis.

To quantify the steady-state variability of nascent proteins per mRNA in units of mature protein (ump), we used a second, independent calibration construct that contains only a single

epitope for FLAG ([6], see Methods) and which we measured using higher laser intensities. After calibration, the number of mature proteins per mRNA was rounded to the nearest integer $d_j$ for a larger number of spots (1844 to 302 spots per frame for 50 imaging frames) for a total of 6435, 3973, and 751 spots for KDM5B, $\beta$-actin and H2B, respectively. The resulting data histograms were down-sampled to create an effective population of 100 translating mRNA spots for each gene, and histograms of these measurements are presented by the black lines in Fig 4A–4C, middle.

We explored if the full stochastic model could be fit to capture simultaneously the experimentally measured steady-state histogram of nascent proteins as well as the temporal dynamics of nascent protein fluctuations on single mRNA. For model comparison to the steady-state histograms, we ran 300 independent simulations per gene and parameter combination ($\Lambda$) and estimated the probability to observe intensities corresponding to $d = 1, 2, \ldots$ mature proteins per mRNA. We denoted resulting probability mass vector as $P(d;\Lambda)$. Assuming that translation on each mRNA is independent of the rest, we could then compute the likelihood of the steady-state intensity data for each gene given the model as:

$$L_{Dist}(Data|Model) = \prod_{j=1}^{100} P(d_j; \Lambda),$$
(24)

and the log-likelihood could be computed:

$$\log L_{Dist}(Data|Model) = \sum_{j=1}^{100} \log P(d_j; \Lambda).$$
(25)

As non-translating spots could not be separated from spots below a basal FLAG intensity in the experimental data measurements, comparison between simulations and measured distributions ignore all spots with an intensity value less than 1/2 ump.

To compare temporal dynamics of the experiments to those of the model, we assumed that errors in the measurement of the average auto-covariances would be approximately normally distributed with variances equal to the measured standard error of the mean [30]. Under this assumption, the probability to measure an auto-covariance of $G_D(\tau_i)$ at lag time $\tau_i$ according to a model that predicts $G_M(\tau_i;\Lambda)$ for parameter set $\Lambda$ is:

$$L_{AC}(G_D|G_M(\Lambda)) = \prod_{i=1}^{N_t} \frac{1}{\sqrt{2\pi\sigma(\tau_i)^2}} \exp\left(-\frac{(G_D(\tau_i) - G_M(\tau_i;\Lambda))^2}{2\sigma(\tau_i)^2}\right),$$
(26)

where $\sigma(\tau_i)$ is approximated by the measured SEM auto-covariance at each $\tau_i$. The logarithm of this likelihood function can then be written as:

$$\log L_{AC}(G_D|G_M(\Lambda)) = C - \sum_{i=1}^{N_t} \frac{(G_D(\tau_i) - G_M(\tau_i;\Lambda))^2}{2\sigma(\tau_i)^2},$$
(27)

where $C$ is a constant that does not depend upon the parameter set $\Lambda$, and the second term is the definition of $\chi^2$ [30] for the comparison of experimental and model-derived autocorrelation analyses.

Because the steady-state distributions and the temporal dynamics were measured using independent experiments, the total likelihood function to match both datasets is the product of the individual functions, and the total log-likelihood is the sum of the individual log-

likelihoods:

$$\log L_{\text{total}}(Dist, G|M) = \sum_g (\log L_{Dist}(Data|Model) + \log L_{AC}(G_D|G_M(\Lambda))), \tag{28}$$

for $g$ = KDM5B, H2B and $\beta$-actin. Now, that we have defined a log-likelihood function to compare the data to the model under different parameter combinations, we can explore parameter space, first to maximize this likelihood and then quantify what is the uncertainty in parameters given the data.

Codon-dependent translation rates were assumed to be consistent among the three genes, as defined in Eq 4, but the three genes were allowed to have different initiation rates, $\{k_i^{(g)}\}$. Under this assumption, the model has a total of four parameters. Upon fitting these parameters to maximize Eq 28, we found that the model could capture both the experimental distributions of nascent proteins per mRNA and the auto-covariance plots for all three genes, as shown in Fig 4A–4C (middle and right). Optimized parameters and their uncertainties (see Methods) were found to be:

$$\begin{cases} \bar{k}_e = 10.6 \pm 0.72 sec^{-1}, \\ k_i^{(\text{KDM5B})} = 0.022 \pm 0.004 sec^{-1}, \\ k_i^{(\beta-\text{actin})} = 0.05 \pm 0.01 sec^{-1}, \\ k_i^{(\text{H2B})} = 0.066 \pm 0.019 sec^{-1}. \end{cases} \tag{29}$$

## Exploring how translation dynamics vary with different parameters

After determining that our model was sufficient to reproduce the experimentally measured fluctuation dynamics for H2B, $\beta$-actin, and KDM5B, we next extended our analyses to consider a broader range of translation parameters. Specifically, we sought to explore the effects of variations to initiation and elongation rates as well as effects of synonymous codon substitutions or modulation of tRNA concentrations.

**Ribosome collisions are rare at most experimentally observed translation initiation and elongation rates.** Previous experimental reports [6–10] estimated a range of values from 0.01 to 0.08 $sec^{-1}$ for the translation initiation rate, $k_i$, and range from 3 to 13 aa/sec for the average elongation rate, $\bar{k}_e$. Using $\beta$-actin gene as a reference, Fig 5A depicts the variation in ribosome density as a function of the base parameters $k_i$ and $\bar{k}_e$, and Fig 5B shows the number of times an average ribosome would collide with an upstream neighboring ribosome during a single round of translation. For most parameter combinations, ribosome loading was predicted to be very low (i.e., fewer than one ribosome per 100 codons) and collisions were rare (i.e., fewer than 10 collisions in an average round of translation). However, for slow elongation and fast initiation, such as those measured by Wang et al. [7]), a ribosome could collide with other ribosomes an average of $\sim$20 times for a gene the length of $\beta$-actin. To further illustrate the effects that these initiation and elongation rates would have on ribosome dynamics on different genes, Fig 5C shows simulated kymographs for SunTag-24X-Kif18b [10], FLAG-10X-KDM5B [6], and SunTag-56X-Ki67 [9], each with their previously reported initiation and elongation rates. In addition, S1 and S2 Figs provide more detailed results of the translation elongation simulations for $\beta$-actin translation at multiple initiation rates and elongation rates, respectively. Each of these kymographs indicates that ribosome dynamics can vary from collision-free dynamics (SunTag-24X-Kif18b and FLAG-10X-KDM5B) to dynamics with multiple collisions (SunTag-56X-Ki67) and that collisions can become more prevalent at high initiation rates or low elongation rates.

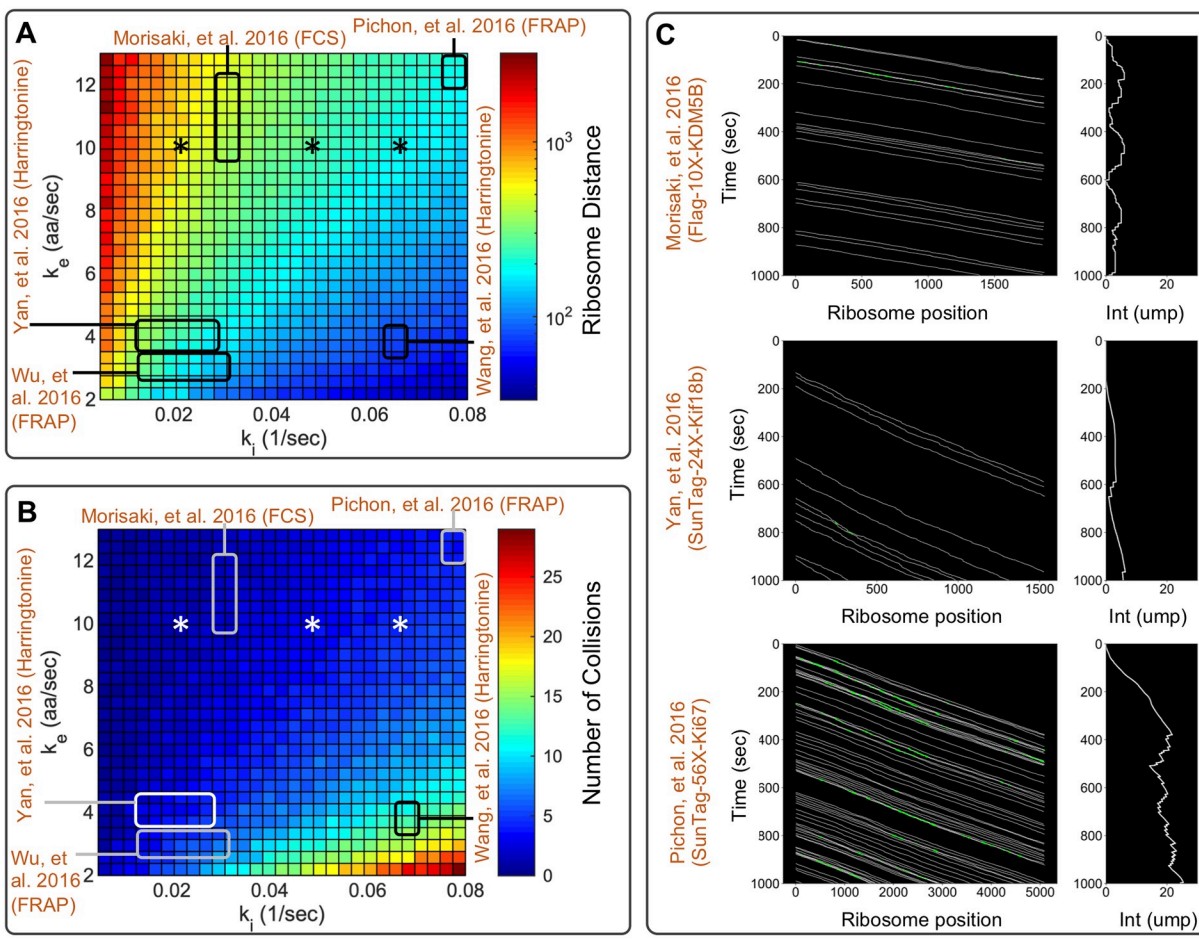

**Fig 5. Ribosome dynamics under experimentally reported initiation and elongation rates.** A) Simulated mean number of codons between ribosomes for the $\beta$-actin gene as function of initiation and elongation constants. In the plot, previous literature initiation and elongation values are highlighted by the squares [6–10], and values estimated in this study are denoted by asterisks. B) Simulated number of collisions per ribosome as a function of initiation and elongation constants. C) Top panel, kymograph showing the ribosomal dynamics for SunTag-24X-Kif18b using experimentally determined parameters $k_i$ = 1/100 sec$^{-1}$ and $\bar{k}_e$ = 3.1 aa/sec [10]. Center panel, kymograph showing the ribosomal dynamics for FLAG-10X-KDM5B using experimentally determined parameters $k_i$ = 1/30 sec$^{-1}$ and $\bar{k}_e$ = 10 aa/sec [6]. Bottom panel, kymograph showing the ribosomal dynamics for SunTag-56X-Ki67 using experimentally determined parameters $k_i$ = 1/13 sec$^{-1}$ and $\bar{k}_e$ = 13.2 aa/sec [9]. White lines in kymographs represent single ribosome positions, and green spots represent ribosome collisions.

**Codon usage affects translation speed and ribosome loading.** Simulations of genes H2B, $\beta$-actin, and KDM5B showed that each gene's codon order influences the overall ribosome traffic dynamics, creating a non-uniform distribution of ribosomes along the mRNA (S3 Fig). This observation of codon dependence led us to look more deeply into possible effects that optimization could have on observable translation dynamics. S4 Fig depicts simulated kymographs for the $\beta$-actin protein for three synonymous sequences containing: (i) natural codons, (ii) most frequent synonymous codon (optimized), and (iii) least frequent synonymous codon (de-optimized). For each case, S4B Fig illustrates the corresponding ribosome loading profiles; S4C Fig shows the simulated distribution of FLAG intensities in units of mature proteins, and S4D Fig presents the corresponding simulated fluorescence auto-covariance functions. S5 and S6 Figs show similar results for the H2B and KDM5B genes, respectively.

In all cases, optimized gene sequences speed-up ribosome dynamics, and de-optimized sequences cause a slower elongation rate that is observed in the auto-covariance plots given in

S4D, S5D and S6D Figs. Moreover, for constant initiation rates, faster elongation would lead to lower ribosome loading (S4B, S5B and S6B Figs) and therefore lower fluorescence intensities, as shown in the distributions given in S4C, S5C and S6C Figs. All three genes under consideration had natural codon usage that was enriched for the most common codons (i.e., the natural and common codon usage dynamics are very similar), such that the translation rate, ribosome loading, and fluorescence intensity could be substantially altered only by substitution to rare codons. We note that the substitution of rare codons would lead to slower elongation and substantially higher numbers of ribosome collisions.

**Depletion of tRNA levels can induce ribosome traffic jams.** In addition to modulating translation speed through codon substitution, it is possible to perturb these dynamics through experimental modulation of tRNA concentrations. For example, Gorgoni et al., [16] used a mutated allele to the gene for $tRNA_{CUG}$ to reduce the concentration of the glutamine tRNA. To study how ribosome dynamics can be affected by the removal or addition of specific tRNA, we simulated the translation dynamics of H2B, $\beta$-actin, and KDM5B at several different concentrations for $tRNA_{CTC}$. S7 Fig shows the effect of decreasing $tRNA_{CTC}$ concentration on the ribosome association time (left) and elongation rate (right). The simulations show that ribosome dynamics are relatively unchanged provided that the $tRNA_{CTC}$ concentration remains above approximately 10% of the native level. In contrast, depleting $tRNA_{CTC}$ concentration below 10% of wild-type levels could lead to ribosome stalling, which was reflected in long ribosome association times and low effective elongation rates. Ribosome traffic-jams are observed under very low $tRNA_{CTC}$ concentration as shown in S8 to S10 Figs. The prevalence of the CTC codon was found to be important in that the effect of $tRNA_{CTC}$ depletion occurs at higher $tRNA_{CTC}$ concentrations for the CTC codon rich KDM5B gene than for the other two constructs.

## RNA sequence to NAscent protein simulation (rSNAPsim)

To facilitate the simulation of single-molecule translation dynamics, all models and analyses described above have been incorporated into a user-friendly Python toolbox, which we have called rSNAPSIM. This toolbox combines a graphical user interface (GUI) divided into multiple tabs, graphical visualizations, and tables to present calculated biophysical parameters (see Fig 6). This simulator performs stochastic simulations considering the widely accepted mechanisms affecting ribosome elongation, such as codon usage and ribosome interference. The toolbox is available in Python 2.7/3.5+ and wrappers for optimized C++ code are provided with installation instructions.

rSNAPSIM takes as an input the gene sequence in Fasta format or an NCBI accession number. The user can decide on the type (FLAG, SunTag, or Hemagglutinin), number, and placement of different epitopes upstream, downstream or within the protein of interest. The toolbox provides the user with a visualization of the gene sequence and the overall gene construct including the position of the POI and the positions of the Tag epitopes. From the concatenated tags and POI sequences, rSNAPSIM automatically generates a discrete single-RNA translation model with single amino acid resolution and codon-dependent translation rates. Once generated, these models can be simulated using stochastic dynamics, and the results can be quantified in terms of predicted translation spot intensity fluctuations (i.e., single-RNA translation time traces or kymographs), ribosomal density profiles, and fluorescence signal auto-covariance. The graphical user interface also provides for easy generation of simulated results for several different experimental assays, including FCS, FRAP, and ROA. From these simulation results, biophysical parameters such as the overall elongation rate or ribosome association rate are automatically calculated and returned to the user. The toolbox provides

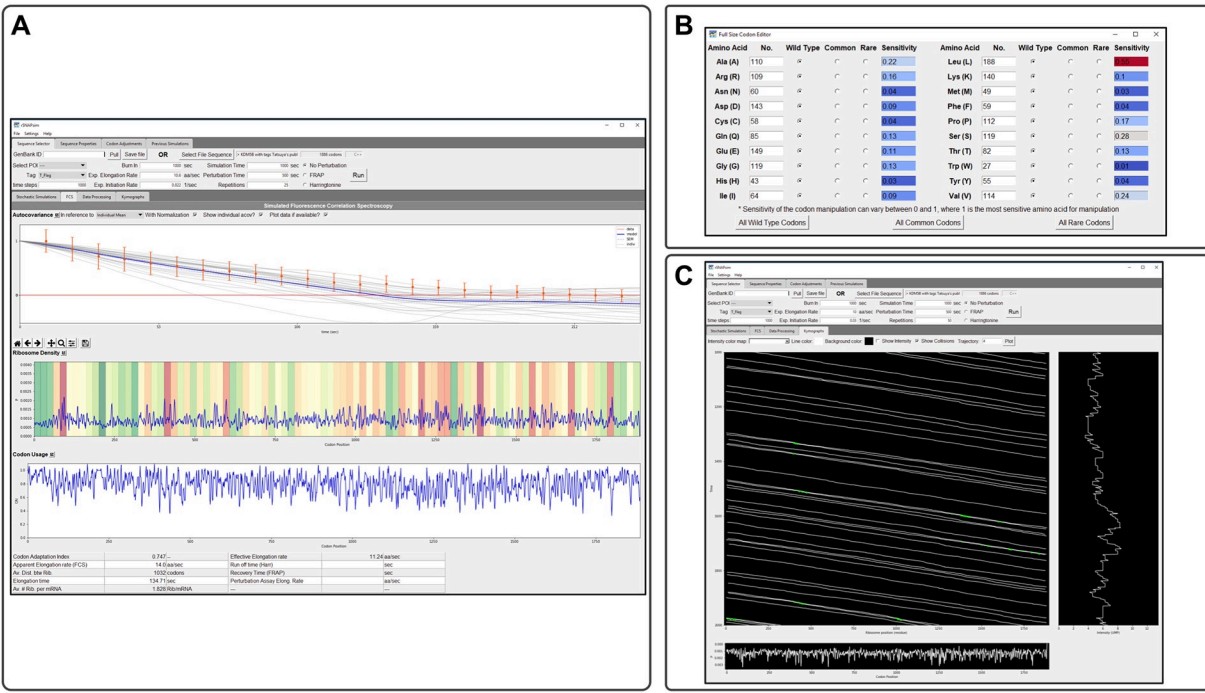

**Fig 6. RNA Sequence to NAscent Protein Simulation (rSNAPsim).** A) RSNAPSIM is divided into four upper tabs and three lower tabs. Upper tabs allow the user to select and adjust sequences and then run simulations under varying conditions. Sequence selector allows the user to load a raw text file or GenBank file for their simulation needs. An option to pull from GenBank via accession number is available. All simulation parameters are also set on this tab. B) After a file is loaded, RSNAPSIM allows the user to change the tRNA copy numbers and codon types under the Codon Adjustment tab. Post simulation, the lower tabs display simulation information such as average intensity over time of N simulations. C) Screen-shot of a kymograph. The kymograph tab allows the user to create their kymographs with varying display options. The Stochastic Simulation tab shows the time course data from the selected simulations. The Fluorescence Correlation Spectroscopy tab displays and compared simulated and experimental single-molecule translation dynamics, the auto-covariance function, and biophysical parameters, such as the elongation constant or ribosomal density. All functionality in the GUI is also available in a command-line module for Python included with RSNAPSIM.

additional interfaces for the user to design and simulate gene sequences with substitution between natural, common, or rare codons for any combination of amino acids or to manually adjust the concentration of tRNA for specific codons. Simulations are saved automatically so that the user can compare translation dynamics for multiple different gene constructs. The toolbox allows for the user to load experimental single-mRNA fluorescence trajectories, compute auto-covariance functions with various normalization assumptions and compare these to model results. For example, the RSNAPSIM screenshot in Fig 6A shows a comparison of model and experimental normalized auto-covariances for KDM5B.

The open-source toolbox was tested in Mac, Windows, and Linux operating systems and is available at: https://github.com/MunskyGroup/rSNAPsim.git. Simulating a gene with 1567 codons for 100 repetitions of 5000 seconds each takes less than 1 minute using a laptop computer with a Core i7 and 32GB of RAM.

## Discussion

Imaging translation in living cells at single-molecule resolution is a new experimental technology that has been applied to only a few genes so far [6–10, 14, 15], but the number of such studies is expected to grow considerably in the near future [12]. Computational models can aid in this research by extracting improved biophysical understanding and parameters from

single-molecule data. For example, in related analyses of transcription dynamics, Rodriguez et al., [18] used a coarse grained stochastic model to capture the polymerase elongation process and reproduce transcription dynamics for a multi-state promoter. Here, we extended that theoretical framework to include the most widely accepted mechanisms affecting nascent protein translation, including codon-dependent elongation and ribosome interference [17] and with specific attention to the placement of fluorescent probes. To complement previous models that have sought to reproduce data from earlier bulk cellular assays [16], and ribosome profiling data [31, 32], our focus has been to integrate single-mRNA stochastic dynamics models with data from *in vivo* single-RNA translation dynamics experiments.

We developed a general codon-dependent model, where nascent protein distributions and auto-covariance functions were generated by detailed stochastic simulations that tracked the positions of ribosomes relative to their neighbors. However, in the absence of perturbations to change initiation and elongation rates, most ribosomes do not encounter others during elongation (Fig 2), at least not at currently accepted elongation and initiation rates from the literature [6–10]. This observation justifies an assumption of sparse ribosome loading and independent ribosome motion, which allow the linear reaction rate reformulation of the codon-dependent translation model into a simplified stochastic moment model and further reduction led to analytical expressions for the steady-state mean and variance of fluorescence in units of mature protein levels per mRNA (Eq 21) and for the decorrelation time (Eq 18). For initiation rates at or below reported experimental values, the simplified analytical model and the full model are in strong agreement (Fig 2). However, increasing initiation rates relative to the base elongation rate, inserting more rare codons into the sequence, or depleting tRNA levels for some codons will increase the number of ribosome collisions and violate the simplifying assumptions (Figs 2C and 5). In such circumstances, the full stochastic model predicts slower effective elongation rates, longer ribosome association times, and accumulation of more ribosomes per mRNA.

With the full and reduced models in hand, it becomes possible to predict how well three modern methodologies would estimate elongation rates from single-molecule measurements: Fluorescence Correlation Spectroscopy (FCS) [12], Fluorescence Recovery After Photobleaching (FRAP) [8, 9, 12], and Run-Off Assays (ROA) after perturbation with inhibitory drugs [7, 10]. Through simulations on 2,647 genes, we demonstrated that estimating elongation rates for long genes (>1000 codons) could be achieved with great accuracy using any of these methodologies, provided that a minimal number of mRNA spots are considered and with an appropriate temporal resolution as demonstrated in Fig 3. However, our results suggest that FCS would be the most likely method to provide an accurate elongation rate estimate (Fig 3A), especially for small and medium size genes. Although our simulation results suggest that FCS is the best single-molecule option to estimate elongation rates, it is important to remark that FCS analysis requires the tracking and measurement of intensity for single spots over long periods of time, and such measurements are susceptible to photobleaching and molecular motion. The former issue has been addressed through the application of optical techniques such as highly inclined thin illumination microscopy [33] and the latter could be addressed through the application of molecular tethers to reduce motion [10]. On the computational side, one could potentially address concerns of bleaching or motion relative to the imaging plane by including hyper-parameters to describe these dynamics and then fit these hyper-parameters concurrently with model parameters using Bayesian analyses.

Run-off assays using Harringtonine to prevent translation initiation can give accurate estimates when genes are long, but the accuracy of such an approach is highly diminished for shorter genes (Fig 3B) or if the precise time of drug action on the mRNA is not known. Our analyses suggest that run-off assays directly depend on the number of ribosomes actively

translating the mRNA at the time of perturbation, and since this number is highly susceptible to stochasticity on small genes, the ROA would require analysis of a much larger number of spots to achieve accurate results.

Our analyses show that FRAP gives poor estimates for all genes of all sizes, and for all tested experimental designs, Fig 3C. The recovery of the intensity after photobleaching depends heavily on the initiation rate, which has been found to be an order of magnitude smaller than the elongation rate, making the recovery a highly stochastic process as well. We directly compared the error size for the studied methods, obtaining that the error in FRAP and ROA is two times larger than in FCS, S11 Fig.

Using FCS data, we demonstrated that a codon-dependent translation model containing one universal average elongation rate and one gene-dependent initiation rate could capture quantitatively the distribution of nascent proteins per actively translating mRNA, as well as the temporal dynamics, for three different genes expressed in human U2OS cells (Fig 4). Combining these estimates of initiation and elongation rates with reported values for the same rates identified using other methods and for other genes, we could predict ribosome dynamics and nascent protein intensities for reported gene sequences [6–10, 14, 15], (Fig 5). Those results allowed us to conclude that relatively fast elongation rates help maintain substantial space between ribosomes on a single mRNA. As a result, these ribosomes should not often collide, and the final ribosome-mRNA association times should remain unchanged for typical initiation rates, natural codon usage, and normal tRNA availability, as shown in S3 Fig. Nevertheless, ribosome dynamics may be affected by genetic or environmental perturbations, such as increased initiation rates (S1 Fig), reduction of elongation rates (S2 Fig), enrichment for rare codons (S4 to S6 Figs), or depletion of tRNA (S7 to S10 Figs).

The present model and RSNAPSIM toolkit have intentionally been made as general and adaptable as possible to efficiently simulate and capture the most accepted mechanisms taking place during translation, i.e. codon-dependent elongation and ribosome interference. At present, the specific rates of codon-dependent elongation are only approximate and based on the prevalence of the corresponding tRNA in the human genome [16]. By modifying this assumption, it is possible to further improve fits for the elongation dynamics shown in Fig 4, and one could find codon dependent rates to explain the diversity of experimentally measured elongation rates depicted in Fig 5. For now, we argue that data from fewer than a dozen genes (and in different cell lines) is as yet insufficient to fully constrain codon dependent rates for all 64 codons. However, as new data is collected for more and more genes, we envision that it will become possible to tune these parameters with greater precision and to capture a greater complement of genes.

In addition to variation in initiation, elongation, codon usage, and tRNA concentrations, many other factors have been described to affect ribosome dynamics. These include, but are not limited to, ribosome stalling or drop-off, pauses due to secondary structures of the specific mRNA, and the electrostatic and hydrophobic interactions between the mRNA and the ribosome [17, 32]. We expect that the increased prevalence of single-RNA translation experiments will add to the current understanding and reveal additional mechanisms taking place during translation. At the same time, such discoveries are bound to create new layers of model complexity. Although these mechanisms have not yet been implemented in our present model, they can be captured easily through modification of the set of elongation parameters, $k_e(i)$. For example, the RSNAPSIM toolbox allows for direct modification of elongation rates at a specific codon, which can be used to mimic pauses at certain locations. Furthermore, all of the computational analyses described above are easily adapted to allow for analysis of simultaneous multi-frame translation dynamics (e.g., when translation occurs on overlapping open reading frames as is the case during frame-shifted translation), as we implemented and described in

[14]. Similarly, the code is easily extended to analyze translation of genes that contain more than one set of fluorescence tags in multiple colors, as has been explored experimentally in [15].

A main limitation in the experimental determination and quantification of translation mechanisms is the specific design of the experiment to make that quantification. For example, in its current form, the introduction of tag regions in the open reading frame of the gene of interest can dramatically alter the overall translation dynamics. As depicted in Fig 1B, the tag region is around 300 codons in length, and this added length can substantially bias the measurement biophysical parameters, especially when quantified using FRAP or run-off assays (see Fig 3). On the one hand, our model can help to explain these differences (S11 Fig), but more importantly, the models themselves can be used to simulate and evaluate different computational designs to determine which are more likely to reveal important biophysical mechanisms or parameters. We envision that user-friendly simulations, such as those provided by RSNAPSIM, can be used to optimize combinations of probe placement, gene length, codon usage differences, video frame rates, drug-based perturbations, or specifications of movie length.

Such simulation-based designs can be conducted prior to any new experimental analysis and then used again to fit the results of those experiments, to pinpoint discrepancies that may reveal new mechanisms, and to refine model parameters and mechanisms. Such integration of experiment and computational model can help set the stage for more efficient experiments that specifically target and quantify the full complement of factors that modulate translation dynamics in living cells.

## Materials and methods

### Studied gene constructs

To constrain our analyses, we use published gene sequences used on single-molecule translation studies. An initial set of sequences were obtained from Morisaki et al., [6], these constructs encode an N-terminal region with 10 repeats of FLAG-SM-tag (318aa) followed by one of three different genes of interest: KDM5B (1549 aa), $\beta$-actin (375 aa) and H2B (128 aa), the 3' UTR region contains 24 repetitions of the MS2 stem-loops. A second source of gene sequences comes from Yan, et al., [10], this gene construct encodes 24 repeats of SunTag followed by the gene of interest kif18b (1800 aa), and the 3' UTR contains 24 repeats of the PP7 bacteriophage coat protein. A sequence encoding 56 SunTag repeats, the gene of interest Ki67 (3177 aa), and the 3' UTR containing 132 repeats of MS2 stem-loops was obtained from Pichon et al., [9]. Finally, multiple gene constructs were build using 10 repeats of FLAG-SM-tag followed by a human gene. The studied human genes come from a comprehensive list of 2,647 gene sequences obtained from the PANTHER database [25].

### Correction to mean and variance of fluorescence intensity for the theoretical model

Neglecting ribosome exclusion, and under an assumption of memory-less initiation with exponential rate $k_i$, the number of ribosomes to initiate translation in a fixed time, $\tau$, is described by a Poisson distribution with mean and variance equal to $k_i\tau$. For a single probe site, we can fix $\tau$ as the time it takes a ribosome to move from that site to the end of the mRNA, and the mean and variance of nascent protein fluorescence can be estimated in terms of units of mature protein fluorescence according to Eq 21.

However, for probes that are spread out across a finite tag region, this distribution requires a slight correction to account for ribosomes within the probe region that only exhibit partial protein fluorescence. Let $\alpha(s)$ denote the intensity, scaled in units of mature protein, exhibited by a ribosome at the position, $s$, along the mRNA as follows:

$$\alpha(s) = \begin{cases} s/L_\mathrm{T} & \text{for } 0 \leq s < L_\mathrm{T} \\ 1 & \text{for } L_\mathrm{T} \leq s \leq L \end{cases} \tag{30}$$

Under an assumption of uniform codon usage, a given ribosome on the mRNA has equal probability to be at any site along the mRNA. If there are an average of $\mu$ mRNA total on the mRNA, then the number at each location is approximated by a Poisson distribution with mean and variance both equal to $\mu/L \cdot ds$. Recall that the mean of the sum of two independent random variables is the sum of two means. Therefore, to find the total mean intensity contribution for all ribosomes on an average mRNA (Eq 22), we can integrate along the length of the mRNA to find:

$$\mu_I = \int_0^L \frac{\mu}{L} \alpha(s) ds, \tag{31}$$

$$= \left(1 - \frac{L_\mathrm{T}}{2L}\right) \mu. \tag{32}$$

Similarly, we recall that the variance of a random variable with variance $\sigma^2$ and scaled by $\alpha$ is equal to $\alpha^2 \sigma^2$ and the variance for the sum of two such variables is the sum of the corresponding variances. Therefor, by noting that $\mu = \sigma^2$, we can find the total variance of intensity on a single mRNA (Eq 23) as:

$$\sigma_I^2 = \int_0^L \frac{\mu}{L} \alpha(s)^2 ds, \tag{33}$$

$$= \left(1 - \frac{2L_\mathrm{T}}{3L}\right) \mu. \tag{34}$$

## Fluorescence Correlation Spectroscopy (FCS)

FCS is usually implemented by computing and comparing the auto-covariances (or autocorrelations) of fluorescence intensities of one or more particles within small fixed volumes [34, 35], but similar correlation analyses have been used to quantify intensity fluctuations for tracked single particles [2]. For our analysis, we compute the temporal auto-covariance times of the FLAG fluorescence signal intensity for a moving volume that is centered around the moving RNA spot.

To estimate the rate of translation elongation, we took the following approach: first, each experimental and simulated intensity time courses were centered to have zero mean by subtracting the average intensity of the time series, and then we normalize with respect to the standard deviation. Next, we computed the covariance function of the fluorescence intensity for each intensity spot according to the standard formula:

$$G(\tau) = \mathbb{E}\{(I_t - \mu_t)(I_{t+\tau} - \mu_{t+\tau})\}, \tag{35}$$

where $\tau$ denotes the time delay and $\mathbb{E}\{v\}$ denotes the expectation of some arbitrary value $v$.

To reduce the effects of high-frequency shot noise and tracking errors that are not considered in the model, the zero-lag covariance $G(0)$ was removed from the analysis [36]. For simulated data, we normalize the auto-covariance function by the simulated variance, $G(0)$, which we can compute directly. For the experimental data, we cannot measure $G(0)$ directly because it is dominated by shot noise, so we instead interpolate $G(0)$ using a linear interpolation of the first four points of the measured auto-covariance function. For statistical purposes, auto-covariances for multiple intensity time courses were calculated, and their value was averaged. Final results are reported as mean values and standard error of the mean (SEM). This signal analysis allowed us to measure the dwell time ($\tau_{FCS}$) at which $G(\tau) = 0$, from which the average ribosome elongation rate can be calculated as:

$$k_e^{(FCS)} = L/\tau_{FCS}. \tag{36}$$

## Parameter uncertainty

Parameter uncertainty analyses were calculated by building parameter distributions that reproduce results within a 10% error, calculated from 1,000 independent simulations using randomly selected parameter values. Simulations were performed on the W. M. Keck High-Performance Computing Cluster at Colorado State University.

## Numerical methods

For solving the model under stochastic dynamics we used the direct method from Gillespie's algorithm [37] coded in Matlab 2018b and Python 2.7. ODE models were solved in Python 2.7.

## Codes and experimental data

All codes and experimental data are available at: https://github.com/MunskyGroup/Aguilera_PLoS_CompBio_2019.git.

## Supporting information

**S1 Fig. Effect of initiation rate on ribosome dynamics.** Translation was simulated using the $\beta$-actin gene, varying initiations rates from 0.03 to 0.6, a constant elongation ($k_e$ = 10 aa/sec), and a ribosomal footprint of 9 codons. Top panels show a kymograph of the ribosome movement. Lower panels show the distribution of collisions for each $k_i$.
(TIF)

**S2 Fig. Effect of elongation rate on ribosome dynamics.** Translation was simulated using the $\beta$-actin gene, varying elongation rates, a constant initiation ($k_i$ = 0.06 sec$^{-1}$), and a ribosomal footprint of 9 codons. Top panels show a kymograph of the ribosome movement. Lower panels show the distribution of collisions per each $k_e$.
(TIF)

**S3 Fig. Codon usage and ribosome occupancy.** Translation was simulated using the a $\beta$-actin gene, varying initiations rates from 0.03 to 0.6, a constant elongation ($k_e$ = 10 aa/sec), and a ribosomal footprint of 9 codons. Top panels show a kymograph of the ribosome movement. Lower panels show the distribution of collisions for each $k_i$.
(TIF)

**S4 Fig. Codon optimization designs for $\beta$-actin.** A) Ribosome dynamics for $\beta$-actin under different codon optimization constructs (natural sequence, using only common codons, and using only rare codons). In the kymographs, white lines represent the ribosome positions,

green spots represent ribosome collisions. The average and standard deviation for the number of collisions is $3.2 \pm 0.9$ for the natural sequence, $2.4 \pm 0.8$ collisions for the optimized sequence (common codons), and $6.9 \pm 1.5$ collisions on the de-optimized sequence (rare codons). B) Ribosome loading for the three codon optimization constructs. D) Auto-covariances calculated for the natural gene sequence, a sequence where all codons are replaced by their most frequent synonymous codon (optimized), and a sequence where all codons are replaced by their less frequent synonymous codon (de-optimized). Simulations were performed using the optimized parameter values given in Eq 29.
(TIF)

**S5 Fig. Codon optimization designs for H2B.** A) Ribosomal dynamics for H2B under different codon optimization constructs (natural sequence, using only common codons, and using only rare codons). In the kymographs, white lines represent the ribosome placement, green spots represent ribosome collisions. The average and standard deviation for the number of collisions is $2.9 \pm 0.7$ for the natural sequence, $2.0 \pm 0.6$ collisions for the optimized sequence (common codons), and $6.0 \pm 1.1$ collisions on the de-optimized sequence (rare codons). B) Ribosome loading for the three codon optimization constructs. D) Auto-covariances calculated for the natural gene sequence, a sequence where all codons are replaced by their most frequent synonymous codon (common), and a sequence where all codons are replaced by their less frequent synonymous codon (rare). Simulations were performed using the optimized parameter values given in Eq 29.
(TIF)

**S6 Fig. Codon optimization designs for KDM5B.** A)Ribosome dynamics for KDM5B under different codon optimization constructs (natural sequence, using only common codons, and using only rare codons). In the kymographs, white lines represent the ribosome placement, green spots represent ribosome collisions. The average and standard deviation for the number of collisions is $4.3 \pm 2.0$ for the natural sequence, $2.8 \pm 1.7$ collisions for the optimized sequence (common codons), and $7.8 \pm 3.1$ collisions on the de-optimized sequence (rare codons). B) Ribosome loading for the three codon optimization constructs. D) Auto-covariances calculated for the natural gene sequence, a sequence where all codons are replaced by their most frequent synonymous codon (common), and a sequence where all codons are replaced by their less frequent synonymous codon (rare). Simulations were performed using the optimized parameter values given in Eq 29.
(TIF)

**S7 Fig. Effects of tRNA depletion on ribosomal dynamics.** A) Three different genes were studied: KDM5B (magenta), $\beta$-actin (cyan) and H2B (orange). Left plot shows the ribosome association time as a function of the tRNA$_{CTC}$ concentration. Right plot, shows the calculated elongation rates estimated by dividing the gene length by the average time needed by the ribosome to complete a round of translation. B) Kymographs show the ribosomal dynamics without depletion (upper panels) and with 99% depletion of tRNA$_{CTC}$ (lower panels). Above the kymographs, the bar represents the studied gene, and the gray area represents the tag region, black lines denote the positions CTC codons. The frequency of the CTC codon is 29 for KDM5B, 8 for $\beta$-actin and 2 for H2B. Simulations were performed using the optimized parameter values given in Eq 29.
(TIF)

**S8 Fig. Depletion of specific tRNA$_{CTC}$ for H2B.** Kymographs (left) show the simulated ribosomal dynamics under different percentages of depletion of tRNA$_{CTC}$. At the top of the kymographs, the bar represents the studied gene, the gray area represents the tag region, and black

lines denote the positions of CTC codons. Histograms (right) show the probability of ribosomal collision. Simulations were performed using the optimized parameter values given in Eq 29.
(TIF)

**S9 Fig. Depletion of specific tRNA$_{CTC}$ for β-actin.** Kymographs (left) show the simulated ribosomal dynamics under different percentages of depletion of tRNA$_{CTC}$. At the top of the kymographs, the bar represents the studied gene, the gray area represents the tag region, and black lines denote the positions of CTC codons. Histograms (right) show the probability of ribosomal collision. Simulations were performed using the optimized parameter values given in Eq 29.
(TIF)

**S10 Fig. Depletion of specific tRNA$_{CTC}$ for KDM5B.** Kymographs (left) show the simulated ribosomal dynamics under different percentages of depletion of tRNA$_{CTC}$. At the top of the kymographs, the bar represents the studied gene, the gray area represents the tag region, and black lines denote the positions of CTC codons. Histograms (right) show the probability of ribosomal collision. Simulations were performed using the optimized parameter values given in Eq 29.
(TIF)

**S11 Fig. Error size for the different methodologies used to calculate elongation rates.** Translation was simulated using the a β-actin gene with the optimized parameter values given in Eq 29. Error bars represent the standard deviation (SD) of the number of repetitions given at the top of each plot. Vertical red lines represent the application of Harringtonine for ROA. Vertical red line represents the time of photobleaching for FRAP.
(TIF)

**S1 Table. Codon usage.** Codon usage table calculated from the *Homo sapiens* genome. Table is computed using 93,487 CDS (Coding DNA Sequence), representing a total of 40,662,582 codons [21].
(PDF)

## Acknowledgments

The authors thank Kenneth Lyon and members of the Munsky lab for their feedback on the presented analyses and for their testing of the RSNAPSIM software.

## Author Contributions

**Conceptualization:** Brian Munsky.

**Data curation:** Luis U. Aguilera, Tatsuya Morisaki.

**Formal analysis:** Luis U. Aguilera, Brian Munsky.

**Funding acquisition:** Timothy J. Stasevich, Brian Munsky.

**Investigation:** Luis U. Aguilera, William Raymond, Zachary R. Fox, Michael May, Elliot Djokic, Tatsuya Morisaki, Brian Munsky.

**Methodology:** Luis U. Aguilera, Brian Munsky.

**Project administration:** Timothy J. Stasevich, Brian Munsky.

**Resources:** Brian Munsky.

**Software:** Luis U. Aguilera, William Raymond, Zachary R. Fox, Brian Munsky.

**Supervision:** Timothy J. Stasevich, Brian Munsky.

**Validation:** Luis U. Aguilera, William Raymond, Brian Munsky.

**Visualization:** Luis U. Aguilera, William Raymond, Brian Munsky.

**Writing – original draft:** Luis U. Aguilera, Brian Munsky.

**Writing – review & editing:** Luis U. Aguilera, Tatsuya Morisaki, Timothy J. Stasevich, Brian Munsky.

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
