## [Decision Letter · Decision Letter 0]

5 Aug 2019

Dear Dr Munsky,

Thank you very much for submitting your manuscript, 'Computational design and interpretation of single-RNA translation experiments', to PLOS Computational Biology. As with all papers submitted to the journal, yours was fully evaluated by the PLOS Computational Biology editorial team, and in this case, by independent peer reviewers. The reviewers appreciated the attention to an important topic but identified some aspects of the manuscript that should be improved.

We would therefore like to ask you to modify the manuscript according to the review recommendations before we can consider your manuscript for acceptance. In particular, careful attention should be payed to delineating where the novelty of the results presented here as compared to those in the prior publications (referee #2, major comment #1). Your revisions should address the specific points made by each reviewer and we encourage you to respond to particular issues Please note while forming your response, if your article is accepted, you may have the opportunity to make the peer review history publicly available. The record will include editor decision letters (with reviews) and your responses to reviewer comments. If eligible, we will contact you to opt in or out.raised.

- Supporting Information uploaded as separate files, titled 'Dataset', 'Figure', 'Table', 'Text', 'Protocol', 'Audio', or 'Video'.

We hope to receive your revised manuscript within the next 30 days. If you anticipate any delay in its return, we ask that you let us know the expected resubmission date by email at ploscompbiol@plos.org.

Sincerely,

Oleg A Igoshin

Associate Editor

PLOS Computational Biology

William Noble

Deputy Editor

PLOS Computational Biology

[LINK]

Reviewer's Responses to Questions

**Comments to the Authors:**

Reviewer #1: In this paper, the authors introduce a sequence-based stochastic model for simulating detailed translation process. Using this model, the authors generate synthetic data to evaluate three types of single-molecule translation experiment (FCS, ROA, and FRAP) and find that FCS is optimal for estimating translation kinetics. By fitting the model to experimental FCS data for three human genes (KDM5B, β-actin, and H2B), the authors capture the nascent protein statistics and temporal dynamics, and characterize how ribosome activities and translation dynamics vary with different kinetics parameters. To facilitate the application of the model, the authors develop an open-source software package, RNA Sequence to NAscent Protein Simulator (rSNAPsim), which allows simulating the single-molecule translation dynamics of any gene.

I found the manuscript very suitable for PCB, in terms of both the subject matter and the methodology. There are only a few small points that I’d like the authors to address before publication:

(1) In the title of the paper, it is not clear what “computational design” means. The term is not used or defined in the rest of the paper. Do the authors mean that they have applied the computational approach to compare three different experimental assays of measuring translation dynamics? Please clarify in the main text or change the title.

(2) Equation 1 of the paper has several problems: (a) the term “w_i(x_2,…,x_nf+i)” should be “w_1(x_1,…,x_nf+1)”; (b) the term “w_i+1(x_i+1,…,x_nf+i+1)” should be “w_i(x_i,…,x_nf+i)”, (c) there should be ellipsis on the left side of “x_i”.

(3) There are multiple language issues in the manuscript. Following is a list of them. The authors and the editorial team will have to make sure all such errors are corrected.

• In the abstract of the paper, “Finding that FCS analyses are optimal for short or long length genes” should be “Finding that FCS analyses are optimal for both short and long length genes”.

• In the abstract of the paper, “we introduce a new open-source software package, … (rSNAPsim) to easily simulate …” should be “we introduce a new open-source software package, … (rSNAPsim), to easily simulate …”.

• In the abstract of the paper, “… to easily simulate the single-molecule dynamics of any gene sequence …” should be “… to easily simulate the single-molecule translation dynamics of any gene sequence …”.

• In line 2 of the paper, “fluorescence time lapse microscopy” should be “time-lapse fluorescence microscopy”.

• In line 19 of the paper, “Yet, despite their overwhelming importance …” should be “Despite their overwhelming importance …”.

• In line 23 of the paper, “Imaging single-molecule transcription was first achieved …” should be “Single-molecule imaging of transcription was first achieved …”.

• In line 43 of the paper, “For example, Morisaki and Stasevich, [12] recently reviewed …” should be “For example, Morisaki and Stasevich recently reviewed … [12]”.

• In line 52 of the paper, “run off assay” should be “run-off assay”.

• In lines 80-81 of the paper, “single-molecule dynamics of any gene” should be “single-molecule translation dynamics of any gene”.

• In lines 145 of the paper, “Nascent Transcript Kinetics” should be “Nascent Translation Kinetics”.

• In lines 281-282 of the paper, “we obtained RMSE_ROA > 2.0 sec_-1 …, Fig 3B” should be “we obtained RMSE_ROA > 2.0 sec_-1 … (Fig 3B)”.

• In line 354 of the paper, the term “effects of parameters” is confusing. Do the authors mean “exploring how ribosome activities and translation dynamics vary with different parameters”? Please consider modifying the subtitle.

• In line 400 of the paper, the term “lower fluorescence intensity distributions” is confusing, since the intensity distribution is a function, not a single value. Please clarify what property of the distribution is lower.

• At the end of line 515, the authors should start a new paragraph for the discussion of the FRAP experiment.

• In line 541 of the paper, “the the corresponding tRNA” should be “the corresponding tRNA”.

• In lines 561-562 of the paper, “all of the computational analyses describe above” should be “all of the computational analyses described above”.

• In line 563 of the paper, the term “multi-frame and multi-color translation” is confusing. Please clarify what it really means.

Reviewer #2: In this manuscript by Aguilera et al., the authors implement a method for analyzing single mRNA translation data using the SunTag/MS2 system. The approach is based on time-lapse imaging of fluorescence which corresponds to the synthesis of nascent protein. Experimentally, the data can be acquired for fluctuation analysis, fluorescence recovery after photobleaching, and run off, all of which are treated in the manuscript. Computationally, the data can be understood using a master equation approach which the authors then solve in a few limiting cases. Finally, they implement an analysis package based on stochastic simulations. This paper does not report any new biological findings per se, and the theoretical advance is minimal if any. They test their computational model on thousands of mRNA in silico and reach the conclusion that fluctuation analysis is the most versatile and accurate analysis tool. In summary, analyzing this type data is certainly nuanced and difficult, and there are already a number of experimental labs working in this area. Providing practitioners a tool to analyze fluorescence trajectories, be they from translation or transcription, solves a problem and makes a novel contribution.

Major Comments:

1. The model for single-molecule translation is the same as the model for single-molecule transcription published recently (PMID: 30554876). While I appreciate that the authors have tailored it to translation and parameterized it for codon usage, the math is the same as that presented in the supplement of that paper. At a minimum, the authors should indicate which equations are the same and which are different. In a related point, some of the early reading is a bit tedious and could be condensed or included in a supplement. For example, the theory section builds to two equations (23, 24) which are quite similar to those reported in Ref. 2.

2. There is real added value in comparing the different techniques for quantifying fluorescence time traces. One thing that wasn’t clear to me: is all the fitting done with the simulated Gillespie approach? It should be. In this era, there is really no point in using approximations when there are so many fast simulators. I imagine that is what is implied by this sentence: “this simulator performs stochastic simulations considering the widely accepted mechanisms ribosome elongation, such as codon usage and ribosome interference.”

3. Why is the autocorrelation always normalized by G(0)? This quantity contains the information about initiation rate. In a related point, why does the raw fluorescence trace have to be normalized by the individual intensity? The fluctuations should be sufficient to back out the occupancy. To be clear, what biophysicists call ‘autocorrelation’ is actually the ‘autocovariance’ and usually contains an un-normalized amplitude.

4. Averaging experimental autocorrelations is not trivial and was discussed in Ref. 35. Specifically, for image-based analysis (as done here), the traces are often short and of variable length and may contain few translation events. The sparseness of the data represents a problem for fluctuation analysis. As such, the arithmetic average is not strictly correct. If the goal is to generate a broadly useful analytic tool, I strongly recommend implementing an averaging scheme which is specific to the translation data in the distributed program.

5. How is translational bursting accommodated in the model?

**Have all data underlying the figures and results presented in the manuscript been provided?**

Reviewer #1: Yes

Reviewer #2: No: image data is not provided

PLOS authors have the option to publish the peer review history of their article (what does this mean?). If published, this will include your full peer review and any attached files.

Reviewer #1: No

Reviewer #2: No

---

## [Editor Report · Decision Letter 1]

19 Sep 2019

Dear Dr Munsky,

We are pleased to inform you that your manuscript 'Computational design and interpretation of single-RNA translation experiments' has been provisionally accepted for publication in PLOS Computational Biology.

In the meantime, please log into Editorial Manager at https://www.editorialmanager.com/pcompbiol/, click the "Update My Information" link at the top of the page, and update your user information to ensure an efficient production and billing process.

One of the goals of PLOS is to make science accessible to educators and the public. PLOS staff issue occasional press releases and make early versions of PLOS Computational Biology articles available to science writers and journalists. PLOS staff also collaborate with Communication and Public Information Offices and would be happy to work with the relevant people at your institution or funding agency. If your institution or funding agency is interested in promoting your findings, please ask them to coordinate their releases with PLOS (contact ploscompbiol@plos.org).

Thank you again for supporting Open Access publishing. We look forward to publishing your paper in PLOS Computational Biology.

Sincerely,

Oleg A Igoshin

Associate Editor

PLOS Computational Biology

William Noble

Deputy Editor

PLOS Computational Biology

---

## [Editor Report · Acceptance letter]

9 Oct 2019

PCOMPBIOL-D-19-01011R1

Computational design and interpretation of single-RNA translation experiments

Dear Dr Munsky,

I am pleased to inform you that your manuscript has been formally accepted for publication in PLOS Computational Biology. Your manuscript is now with our production department and you will be notified of the publication date in due course.

With kind regards,

Bailey Hanna
